# Crystal structure of the plant symporter STP10 illuminates sugar uptake mechanism in monosaccharide transporter superfamily

Peter Aasted Paulsen[1], Tânia F. Custódio[1] & Bjørn Panyella Pedersen [1,2]

Plants are dependent on controlled sugar uptake for correct organ development and sugar storage, and apoplastic sugar depletion is a defense strategy against microbial infections like rust and mildew. Uptake of glucose and other monosaccharides is mediated by Sugar Transport Proteins, proton-coupled symporters from the Monosaccharide Transporter (MST) superfamily. We present the 2.4 Å structure of *Arabidopsis thaliana* high affinity sugar transport protein, STP10, with glucose bound. The structure explains high affinity sugar recognition and suggests a proton donor/acceptor pair that links sugar transport to proton translocation. It contains a Lid domain, conserved in all STPs, that locks the mobile trans-membrane domains through a disulfide bridge, and creates a protected environment which allows efficient coupling of the proton gradient to drive sugar uptake. The STP10 structure illuminates fundamental principles of sugar transport in the MST superfamily with implications for both plant antimicrobial defense, organ development and sugar storage.

[1] Department of Molecular Biology and Genetics, Aarhus University, Gustav Wieds Vej 10, DK-8000 Aarhus C, Denmark. [2] Aarhus Institute of Advanced Studies, Aarhus University, Høegh-Guldbergs Gade 6B, DK-8000 Aarhus C, Denmark. Correspondence and requests for materials should be addressed to B.P.P. (email: bpp@mbg.au.dk)

For all life, sugars play an essential role as both nutrients and signaling molecules. In plants, photosynthetically synthesized sugar is distributed mainly as sucrose throughout the plant body via the phloem. A key two-step process in apoplastic sugar import from the phloem is the breakdown of sucrose by invertases to glucose and fructose, followed by transmembrane uptake into sink cells mediated by Sugar Transport Proteins[1,2]. This tightly regulated process is key for correct development of plant organs like root tips, pollen and seeds, and enables storage of high amounts of soluble sugars in sink tissues such as fruit[1,3,4]. Furthermore, apoplastic sugar depletion through STPs, where sugar is removed from the extracellular space, has recently been identified as a defense strategy against microbial infection including rust and powdery mildew[5–10]. By removing apoplastic sugar, the plant restricts the amount of nutrition available to the pathogen. STPs have also been implicated in nitrogen use and in programmed cell death[11,12]. The large Monosaccharide Transporter (MST) superfamily is responsible for the selective transport of monosaccharides and polyols throughout the plant kingdom[1,13,14]. The MST superfamily structurally belongs to the ubiquitous Major Facilitator Superfamily (87+ protein families in all kingdoms of life)[15,16]. Six different Major Facilitator families have so far been structurally characterized, and not surprisingly all display significant variations in their detailed transport mechanism in line with substrate differences and physiological function[15,17–23]. Fifty three members of the Monosaccharide Transporter Superfamily have been identified in *Arabidopsis thaliana* alone, of which 14 constitute the Sugar Transport Protein family[1,14] (Supplementary Fig. 1 and Supplementary Table 1). STPs display significantly higher sugar affinity compared to most other sugar Major facilitators (up to 1000× fold), and have a broad pH optimum compared to their bacterial counterparts[17,19,24,25]. STPs face the apoplastic space where pH is alkalized as a central stress response to e.g., microbial infection, drought and high salinity[8–10,26,27]. The functional effects of this extracellular alkalization are not well understood, but the phenomenon, which can last from hours to days, is thought to form part of a central plant response to stressors[26]. *Arabidopsis thaliana* STP10 is a recently characterized member of the STP family with classic STP traits. Found in growing pollen tubes, it is a proton driven symporter that displays low μM range affinity for glucose and can transport glucose, galactose and mannose[25]. While being extensively studied, the mechanism behind high affinity substrate recognition and transport in STPs is not understood. Although structures exist of other sugar/H$^+$ symporters and sugar facilitators from other kingdoms of life, they have low sequence identity to STPs and cannot explain the key characteristics of STP transport (Supplementary Fig. 2 and Supplementary Table 2)[17–19,28]. Therefore, we have determined the crystal structure of STP10 with glucose bound in a central binding site. The structure reveals specific interactions mediating high affinity sugar recognition partly through a hydrophobic patch and suggest a proton donor/acceptor pair to connect sugar transport to proton translocation. Towards the extracellular side, a Lid domain, conserved in all STPs, separates both the sugar binding site and the proton binding site from the extracellular lumen. The structure together with biochemical data suggest that the Lid domain creates a protected environment for the proton donor/acceptor pair which efficiently couples the proton gradient to sugar translocation.

## Results

### Crystal structure of STP10.
We have overexpressed STP10 in *Saccharomyces cerevisiae*, biochemically characterized it, and solved the structure. The structure was determined to 2.4 Å resolution using X-ray crystallography, and the final model was refined to an Rfree of 26.8% and includes residues 21-507 (of 514

residues total) (Fig. 1, Supplementary Fig. 3 and Supplementary Table 3). The map has excellent density for the entire model except a single extracellular loop of seven residues and revealed several additional molecules, as well as tightly bound waters (Supplementary Figs. 3 and 4). The asymmetric unit comprises a single monomer with no higher oligomeric state observed, despite STPs possibly being oligomers in a physiological context[6]. The overall structure adopts a Major Facilitator fold with 12 transmembrane helices (M1-M12) divided in two domains (N and C domain) with a quasi-twofold symmetry perpendicular to the membrane plane (Fig. 1a). These are joined by an intracellular helical bundle (ICH) domain. The central transmembrane binding site is located between the N and C domain and contains unambiguous density for glucose (Fig. 1b, c, Supplementary Fig. 4). Towards the extracellular side an unexpected feature emerged as a "helix-helix-loop-helix" domain that we have dubbed the Lid domain due to its resemblance to a small lid that sits over the expected extracellular entry pathway to the sugar binding site. The Lid domain, a fully conserved feature found in all STP sequences, is a protrusion of the first extracellular loop (M1 to M2) and it is covalently linked to the C domain by a disulfide bridge (Cys77 to Cys449) (Fig. 1a and Supplementary Fig. 4). This disulfide bridge locks the N and C domain together at the extracellular side between M2 and M11 in a way that has never been observed before in any Major Facilitator.

STP10 is in a substrate bound outward occluded state with the N and C domains as a clamp around the central binding site. Exit towards the cytosol is blocked, facilitated by several strong interactions between the N and C domain, that has also been observed in other sugar transporters (Supplementary Fig. 5). At the cytosolic side, the ICH domain forms part of this interaction and contributes with several hydrogen bonds to a network that involved both the N, C and ICH domains. This network has strong similarity to the ones described for the sugar transporters XylE and GLUT1, and key residues have previously been implicated in both activity and trafficking of STPs[5,6,28–30]. Towards the extracellular side glucose access is blocked mainly by the Lid domain (Fig. 2a). Glucose is located in the central binding site with well-defined interactions (Fig. 1b, c). The C domain creates a T-shaped CH-π interaction from Phe401(M10) to the main ring of glucose, while a large number of key polar interactions mediate specificity. Asn332(M10) is in contact with the hydroxyl group of glucose carbon 6 (C6) while a hydrogen bond network made through Gln295(M7), Gln296(M7), Asn301 (M7), Asn433(M11), Thr437(M11), Trp410(M10) and the main chain carbonyl of Gly406(M10) and water, mediate the contact to the C1-C4 hydroxyl groups of glucose. From the N domain, only a single polar interaction is observed, from Gln177(M5) to the C1 hydroxyl group and the pyranosyl oxygen. STP10 display high affinity transport of glucose with a Km of 2.6 μM (Fig. 2b and Supplementary Fig. 6a), in good accordance with a previously reported value of 7.6 μM[25]. The apparent Kd of glucose binding to STP10, as determined by isothermal titration calorimetry on the purified sample, is in the same range as the Km, as has been observed previously for other sugar transporters[19] (Fig. 2c). Competition assays using radioactively labeled glucose confirm galactose and mannose as potential substrates (Fig. 2d). Furthermore, growth complementation assays show uptake of glucose at lower sugar concentration (~1 mM), and for both mannose and fructose at higher sugar concentrations (>10 mM) (Supplementary Fig. 6b). High glucose concentrations (>10 mM) appear to inhibit yeast growth as also reported previously for other STPs in growth complementation assays[27,31]. Further growth competition assays confirm the interactions of the polar and CH–π interactions from the C domain (Supplementary Fig. 6c, d). The F401A mutant abolishes transport, highlighting the pivotal

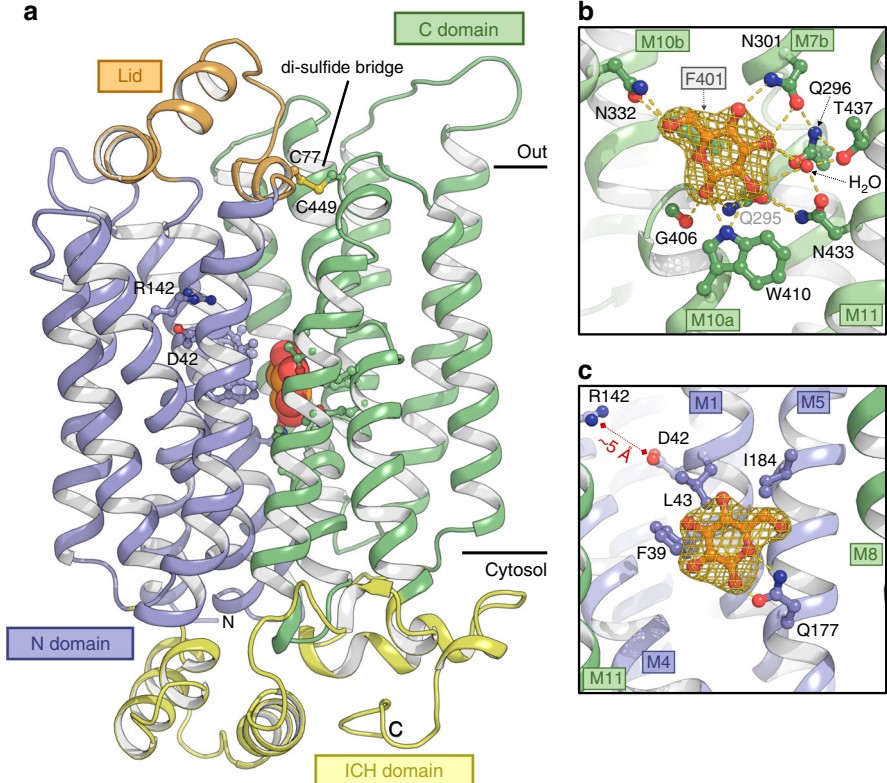

**Fig. 1** Structure of the high affinity Sugar Transport Protein STP10. **a** The structure represents an outward facing occluded state of the sugar transporter in complex with glucose. Glucose (shown as spheres) is buried in the membrane at the interface between the N domain (blue) and C domain (green). Selected residues are shown as sticks. Black bars depict the approximate location of the membrane. **b** The glucose binding site towards the C domain. Yellow dashes indicate hydrogen bonds (2.6–3.6 Å distances) to glucose. The omit mFobs-DFcalc density for glucose is contoured in gold (5σ). **c** Same as panel b for the glucose binding site towards the N domain

function of a CH–π interaction for protein-monosaccharide recognition[32]. As expected, removal of single polar interactions (Q295A, N301A, and N332A) does not completely abolish transport, but appear to give STP10 much lower affinity for its substrate as demonstrated for Q295A (Supplementary Fig. 6c, d). All of these interactions between the C domain and the substrate are also found in bacterial and human sugar transporters (Supplementary Fig. 7). Transport is dependent on the proton gradient as demonstrated by the use of the proton gradient decoupler Carbonyl cyanide m-chlorophenyl hydrazone (CCCP). The protein can transport the non-metabolized glucose analog 2-deoxyglucose, and appear to be sensitive to only some of the inhibitors known from bacterial and human sugar transport (Supplementary Fig. 6e, f).

**Substrate affinity is linked to proton donor/acceptor pair**. The µM affinity of STP10 for glucose can be explained by the sub-strate's interaction to residues in the N domain (Phe39 (M1b), Ile184 (M5) and in particular Leu43 (M1b)) that creates a hydrophobic interaction surface for the substrate (Fig. 1c). This tight and hydrophobic interaction surface is not found in human sugar facilitators or bacterial sugar/H+ symporters, where the interaction distance is longer and the corresponding residue is polar[17–19] (Supplementary Fig. 7). Using tight hydrophobic interactions to boost affinity is a common theme for high affinity protein-ligand and protein-protein complexes[33,34].

A solvent accessible and electronegative cavity below the Lid domain allows contact between the substrate binding site and the core of the N domain (Fig. 2a and Fig. 3a, b). Here we find the only

two buried charged residues in the transmembrane region, Asp42 (M1b) and Arg142(M4) (Fig. 1a, c, and Supplementary Fig. 8). These are the sole candidates for the proton donor/acceptor pair needed for proton translocation[35]. This key role is supported by mutating either Asp42 or Arg142 to alanine which abolishes transport. Arg142 seems to be somewhat more resilient to change as a mutation to lysine does still allow for minimal transport (Supplementary Fig. 6b). The position is similar to the position of proton donor/acceptor pairs in other Major Facilitator proton driven symporters like the bacterial xylose/H+ symporter XylE[36], and the glucose/H+ symporter GlcPse[17] (both ~27% sequence ID to STP10). Interestingly, wheat and barley resistance towards fungal pathogens can be pinpointed to a glycine-to-arginine mutation in exactly this part of the N domain in a wheat sugar transporter (*Lr67res*) and the barley transporter HvSTP13, highlighting the importance of flexibility and charge distribution in this region[6,7]. Asp42 is located on the M1b helix flanked by 6 glycine residues (conserved in all STPs), giving M1b high flexibility, and we propose that local movements of M1b can be controlled by the protonation state of Asp42 (Fig. 3b). The distance between Asp42 and Arg142 is ~5 Å indicating that the aspartate is in a protonated state (Fig. 1c). This is consistent with the low pH of the crystallization condition of 4.5, given that the Asp42 pKa would be expected to increase when removed from the positive charge of Arg142 while buried in a hydrophobic environment[35,37]. It also matches previous observa-tions from other proton symporters and proton pumps[35,36,38]. M1b creates coupling between the protonation and substrate binding sites, as the repulsion of the protonated Asp42 away from Arg142 leads to a visible distortion of this flexible helix towards the substrate binding site. This creates the hydrophobic interaction

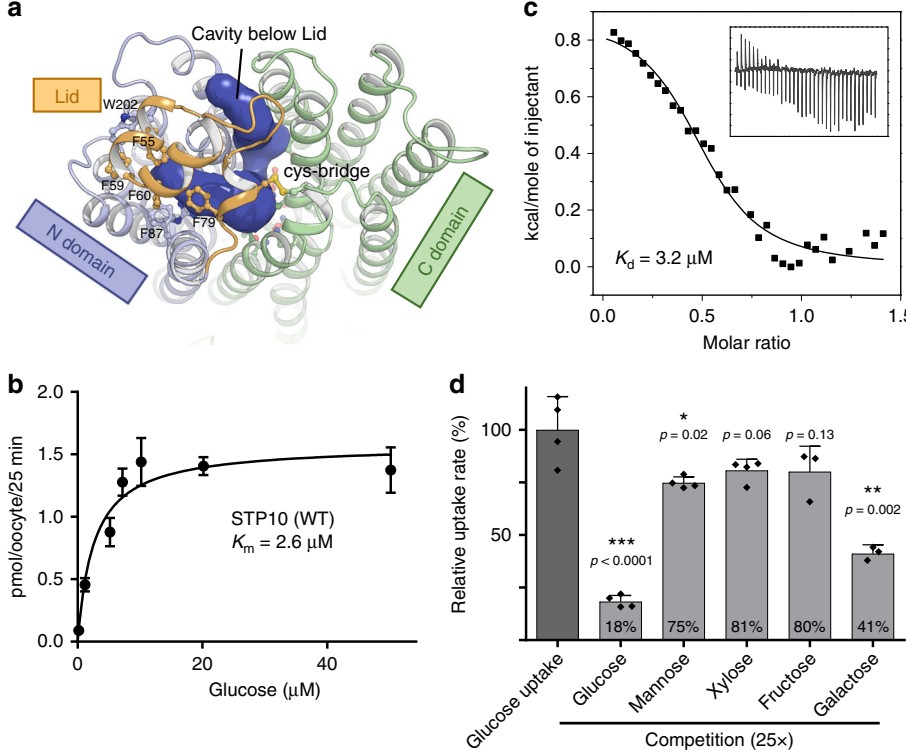

**Fig. 2** Functional characterization of STP10. **a** Glucose access from the extracellular side is blocked by the Lid domain covalently linked to the C domain. **b** Michaelis-Menten fit to glucose titration of STP10 using a *Xenopus* oocyte uptake assay at pH 5.0. **c** Binding affinity between glucose and STP10 by Isothermal titration calorimetry at pH 5.5. **d** Substrate specificity determined by competition in a yeast uptake assay at pH 5.0. *$P \leq 0.05$; **$P \leq 0.01$; and ***$P \leq 0.001$ by Student's *t* test. Data for all assays are mean ± SD of three or more replicate experiments

surface, as defined by Phe39 and Leu43, that closes in towards the glucose molecule (Supplementary Fig. 7). This mechanism is supported by the L43A mutant, which greatly reduces STP10 affinity for glucose and turns STP10 into a low affinity transporter (Km 391 μM) (Fig. 3c). A corresponding mutant L43N which replaces the hydrophobic interaction surface to a polar one of similar size, also leads to a significant decrease (Km 149 μM), highlighting that the hydrophobic aspect of the interaction is a key contributor to affinity (Supplementary Fig. 9a). Supporting this key role of Leu43, a similar pattern is observed in related sugar transporters HUP1 and HUP2 from the algae *Parachlorella kessleri*, where mutating this position changes affinity 20-fold depending on side chain hydrophobicity (Supplementary Fig. 1)[39]. This M1b-linked mechanism to control affinity thus appear to be conserved not only in STPs but also in closely related protein families outside the plant kingdom (Supplementary Fig. 1).

**The Lid domain and the disulfide bridge.** The Lid domain contains a conspicuous cluster of aromatic residues (Phe55(L), Phe59(L), Phe60(L), and Phe79(L), as well as Phe87(M2) and Trp202 (M6)) that isolate the proton donor/acceptor pair from the extracellular space (Fig. 2a and Fig. 3b). These residues are perfectly conserved in all STPs (Fig. 3b and Supplementary Fig. 1). The structure suggests that the Lid domain, when clamped down by the C domain through the disulfide bridge, will help maintain protonation of the Asp42 during transport. To test effects of the Lid domain on Asp42 protonation we mutated the disulfide bridge residues to create a detached Lid domain. At the high substrate concentrations used in the growth complementation assay cell viability appear virtually unchanged in all settings (Supplementary Fig. 6b). However, more detailed investigation

shows that both the Cys77Ala and Cys449Ala mutant becomes increasingly sensitive to alkaline pH and can only function fully at acidic pH (pH < 5) (Fig. 3d and Supplementary Fig. 9). In confirmation of this, only the wt protein is sensitive to reducing agents in an in vivo uptake assay, indicating that the disulfide bridge is present and activity is lowered when the bridge is reduced (Supplementary Fig. 9b). Mutating the equivalent of Cys77 in the wheat gene *Lr67* reintroduces pathogen susceptibility to the resistant gene-version (*Lr67res*)[6], highlighting the impact of this cysteine. Increasing the Lid domain flexibility by removing the disulfide bridge is linked to protonation and does not directly change affinity towards glucose as demonstrated here at low pH. However, at higher pH, Asp42 does not become protonated easily in the Lid mutant, leading to lower turnover and a threefold higher Km (Supplementary Fig. 9c). By breaking the disulfide bridge and increasing flexibility, the proton/donor acceptor pair becomes much more sensitive to the extracellular pH, either directly through a change in Asp42 pKa value or through a requirement for a stronger proton gradient to drive transport, and this indirectly affect substrate turnover at higher pH, without affecting Kd (Supplementary Fig. 9).

## Discussion
Based on these findings, we suggest a model for sugar transport by Sugar Transport Proteins and the Monosaccharide Transporter superfamily (Fig. 4). The Lid domain locks the two transmembrane domains together at the extracellular side via the disulfide bridge. Rearrangements of the N domain and the Lid domain must occur to allow the monosaccharide substrate to bind, but there is no clear entry pathway as seen in other Major Facilitators[18]. However, smaller rearrangements of M1, M5, M8

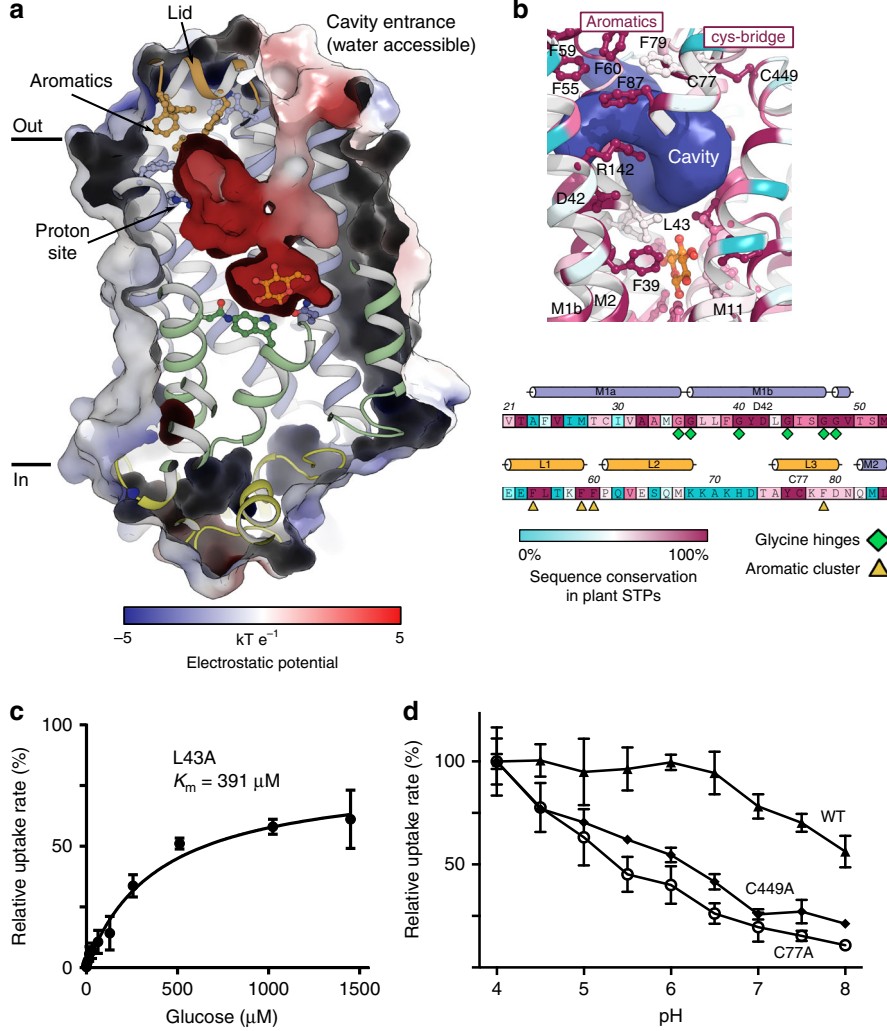

**Fig. 3** The Lid domain and its effect on transport. **a** Electrostatic surface representation showing the negative cavity that connects the proton donor/acceptor site with the glucose binding site. A cluster of aromatic residues on the Lid domain isolate the proton site from the extracellular side. **b** Conservation of the aromatic residues of the Lid domain and residues of the proton donor/acceptor site and M1b. Both structure and sequence is colored according to sequence conservation between 1336 unique STPs (35–95% seq. ID) found across plant species. **c** Michaelis-Menten fit to glucose titration of STP10 mutant L43A at pH 4.0. **d** Glucose uptake rate as determined by a yeast uptake assay at different pH for WT STP10 and mutants C77A and C449A. Data for all assays are mean ± SD of three or more replicate experiments

and the loop region of the lid domain could create an entry pathway to the central binding site (Fig. 3a and Supplementary Fig. 7). The aromatic cluster of the Lid domain isolate the protonation site and enable efficient transport at the physiological pH of the apoplast (around pH 5–6)[26]. The protonation of Asp42 leads to a displacement away from Arg142 and a movement of the flexible M1b helix with Phe39 and Leu43 coming towards the substrate and locking it in. The N domain creates affinity together with the Lid domain, while the polar C domain interactions recognize the specific hydroxyl groups of the substrate and thus can mediate specificity. It remains to be elucidated how substrate release can be achieved, and it is difficult to visualize how the Lid domain will move to accommodate a cytosolic exit pathway. Morphs using inward facing Major Facilitator structures result in serious clashes of the Lid domain with the C domain. The pH tolerance created by the Lid domain can be related to the physiological function of STPs, where STP activity is preserved during stress-induced alkalization of the apoplast. Together with their high affinity for sugars, this will allow local apoplastic sugar deprivation to protect from microbial infections[5–9,27].

In summary we present the structure of an STP protein, highlighting several features conserved in the Sugar Transport Protein family and the Monosaccharide Transporter superfamily. In particular the structure provides an explanation for high sugar affinity, and suggests a mechanism to couple the proton-motive force to sugar transport. A completely unexpected finding is the Lid domain which implies a reevaluation of mobility and the model of transport compared to other Major Facilitators. The structure provides a template for modeling STP and MST proteins that are key regulators of plant development and essential for microbial defense and nutrient uptake in sink tissues throughout the plant. It sheds light on sugar recognition and in particular explain how high affinity sugar transport can be generated, in a process that is essential to all plant life.

## Methods

**Protein purification**. The gene encoding the *Arabidopsis thaliana* protein STP10 (Accession number Q9LT15 [https://www.uniprot.org/uniprot/Q9LT15]) was introduced into an expression construct based on p423_GAL1[40] with a C-terminal purification tag containing a thrombin cleavage site and a deca-histidine tag.

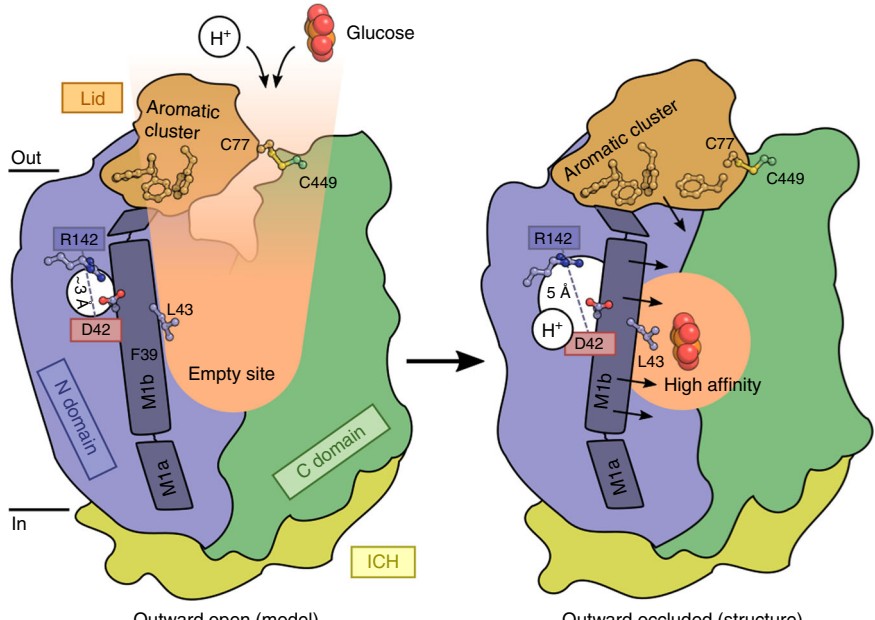

**Fig. 4** Proposed mechanism of glucose coupling to proton donor/acceptor site. In the outward open conformation (left), protons and glucose enter the central binding sites through small rearrangements of the N domain and the Lid domain that is covalently linked to the C domain through Cys77-Cys449. Protonation of Asp42 leads to its repulsion away from Arg142 and pushes the flexible M1b towards the glucose binding site, giving preference to high affinity glucose binding through Phe39 and Leu43 (right, observed structure). The aromatic cluster of the lid helps to isolate the proton donor/acceptor pair and maintain pKa values of Asp42 conductive to transport at a broad range of pH values

The primers used were Fw (GAAAAAACCCCGGATTCTAGAACTAGTGGATC CTCCATGGGTATGGCTGCAGGAGGAGCTTTTG) and Rv (TCCGCCGCTAC CGCCTCCTCCACTACCTCTTGGGACTAGCCCTTAATTGGTATTGTTGTCA TCATGTC). Transformed *Saccharomyces cerevisiae* (strain DSY-5, vendor Gentaur cat# P04003) were grown in a culture vessel to high density by fed-batch and harvested after a 22 h induction using galactose[41]. Harvested cells were washed in cold water, spun down and re-suspended in lysis buffer (100 mM Tris pH 7.5, 600 mM NaCl, 1.2 mM phenylmethylsulphonyl fluoride (PMSF)), followed by lysing using bead beating with 0.5 mm glass beads. The homogenate was centrifuged for 20 min at $5000 \times g$, followed by sedimentation of membranes by ultracentrifugation at $200{,}000 \times g$ for 2 h. Membrane pellets were re-suspended in membrane buffer (50 mM Tris pH 7.5, 500 mM NaCl, 20% glycerol) before being frozen in liquid nitrogen in 3 g aliquots. Six grams of frozen membranes were solubilized for 30 min in a solubilization buffer (150 mM NaCl, 50 mM Tris pH 7.5, 5% Glycerol, 50 mM D-glucose, 1% n-dodecyl-β-d-maltoside (DDM), and 0.1% Cholesterol hemi succinate (CHS)) in a total volume of 100 ml, after which unsolubilized material was removed by filtration using a 1.2 μm filter. Twenty millimolar imidazole pH 7.5 was added and the solubilized membranes were loaded on a pre-equilibrated 5 ml Ni-NTA column (GE Healthcare) at 3 ml/min. After loading, the column was washed with 10 column volumes of W60 buffer (Solubilization buffer with 0.1% DDM and supplemented with 60 mM Imidazole pH 7.5), followed by a 20 column volumes wash with G-buffer (20 mM Mops pH 7.5, 250 mM NaCl, 10% Glycerol, 0.12% Octyl Glucose Neopentyl Glycol (OG-NG), 0.012% CHS, 0.5 mM tris(2-carboxyethyl)phosphine (TCEP)). The composition of the G buffer was optimized through a thermostability assay[42]. The protein was eluted from the column by circulating 5 ml G-buffer supplemented with bovine thrombin and 20 mM Imidazole pH 7.5, at 19 °C for ~16 h. The following day the column was washed with 15 ml of G-Buffer supplemented with 40 mM imidazole. The samples were pooled and concentrated using a spin column (50 kDa cut-off, Vivaspin) to a volume of ~400 μl and injected on a size-exclusion column (Enrich 650, Biorad), pre-equilibrated in G-buffer. Peak fractions were concentrated to ~15 mg/ml and used directly for crystallographic experiments.

**Crystallization**. STP10 was crystallized in lipidic cubic phase (LCP). To prepare lipidic cubic phase for crystallization trials, the protein was supplemented with 100 mM D-glucose before mixing with a 80% monoolein (Sigma-Aldrich) 20% cholesterol mixture, in 1:1.5 protein to lipid/cholesterol ratio (w/w) using a syringe lipid mixer. For crystallization, 50 nl of the meso phase was mixed with 1000 nl of crystallization buffer for each condition on glass sandwich plates using a Gryphon robot (Art Robbins Instruments). Tiny crystals appeared after one day at 20 °C. These crystals diffracted to ~10 Å at Diamond Light Source beamline I24. The addition of various additives and detergents were used to optimize crystals and the final optimized crystallization screen contained 0.1 M NaCitrate pH 4.5, Ammonium dihydrogen phosphate (75–150 mM), DMSO (5–12%), and PEG400 from

25–35%. This gave crystals with a size of approximately $70 \times 10 \times 30$ μm. The crystals were collected using dual thickness micromounts (MiTeGen) and immediately flash frozen in liquid nitrogen. The final datasets were collected at Diamond Light Source beamline I24 using a wavelength of 0.9686 Å.

**Data processing**. Datasets were processed and scaled using XDS[43] in space group $P 2_1$ (#4), which suggested the presence of one STP10 monomer in the asymmetric unit (~54% solvent content). Two datasets derived from two crystals (same drop) were merged to yield the final dataset (Supplementary Table 3). To solve the phase problem, a library of 60 search models was generated and a systematic search of the library and other parameters was done with the MRPM strategy (240 total searches)[44,45]. This identified a Memoir-based[46] and manually pruned homology model of STP10 based on XylE (pdb 4GC0) as the most suitable search model, and a final Molecular Replacement search was done in Phaser[47] with a 3.5 Å cutoff resulting in a solution with TFZ = 6.1. The resulting electron density map was of very low quality with significant model bias, but allowed for the manual adjustment of 10 out of 12 transmembrane alpha-helices at low resolution. Refinement could not proceed with this model. The model was then significantly improved by a combination of Rosetta optimization in phenix. rosetta_refine[48] and Molecular Dynamics based geometry optimization using MDFF[49] through an in-house pipeline tool, Namdinator[50]. After this the model could be successfully subjected to phenix.autobuild[51] and resulted in a model with Rfree of 39%. From here the electron density map allowed for iterative model building in COOT[52] and refinement using phenix.refine[53] guided by 2mFo-DFc maps and Feature Enhanced Maps[54] using model phases. Final refinement in phenix.refine was done with a refinement strategy of individual sites, individual ADP, and group TLS (3 groups), against a maximum likelihood (ML) target with reflections in the 63–2.4 Å range. The final model resulted in electron density maps of excellent quality, and yielded an Rwork of 20.3% and an Rfree of 26.8% (Supplementary Table 3). MolProbity[55] evaluation of the Ramachandran plot gave 95.7% in favored regions and 0.0% outliers. The cavity next to the glucose was identified with CAVER[56] using default settings and a probe radius of 1.4 Å, which is equivalent to the radius of water. All structural figures were prepared using PyMOL (The PyMOL Molecular Graphics System, Version 1.5.0.4 (Schrödinger LLC, 2012)). Conservation of residues across species was analyzed using Consurf[57]. Sequence alignments were constructed with PROMALS3D[58], followed by manually refining gaps based on the transmembrane regions observed in the STP10 structure and predicted for the other sequences using Phobius[59]. Alignments were visualized using ALINE[60].

***Xenopus* oocyte uptake assay**. The atSTP10 gene was subcloned into the EcoRI and NotI sites of the pXOOM plasmid[61]. For cRNA preparation, plasmids were linearized with NheI and the RNA was synthesized using the mMESSAGE mMACHINE T7 Transcription Kit (ThermoFisher). Oocytes from *Xenopus laevis*

were purchased from EcoCyte Bioscience (Castrop-Rauxel, Germany). For expression in oocytes, ~25 ng of RNA produced in vitro was injected into oocytes, using a Nanoject III (Drummond scientific, Broomall, PA). Oocytes were incubated at 18 °C for 2–3 days before measuring transport uptake. Uptake assays were performed as previously described with few modifications[62]. Briefly, groups of 5 oocytes were pre-incubated in Kulori buffer solution pH 5.0 (90 mM NaCl, 1 mM KCl, 1 mM CaCl$_2$, 1 mM MgCl$_2$, 5 mM MES) for 5 min. The pre-incubation buffer was aspirated and replaced with 200 μl of the reaction buffer consisting of Kulori buffer pH 5.0 with 1 μCi [3 H]-D-glucose (PerkinElmer, USA) and 0–100 μM D-glucose (Sigma-Aldrich). The assays were performed in a SpectraPlate-96MB (PerkinElmer, USA) and for each reaction, oocytes were incubated for 25 min at room temperature. The reaction was stopped by aspiration of the reaction buffer and immediate application of ~400 μl of ice cold kulori buffer. The oocytes were further washed four times and transferred individually to a 3 ml scintillation vial. The cells were disrupted by adding 100 μl of a 10% SDS solution followed by immediate vortexing and the addition of 3 ml of EcoScintTM H scintillation fluid (National Diagnostics). The sample radioactivity was quantified by liquid scintillation counting. Data was analyzed with Graph Pad Prism 7. The experiments were performed at least in triplicate and showed similar results.

**Isothermal titration calorimetry**. ITC titrations were performed with a Micro-Cal™ VP-iTC isothermal Titration Calorimeter (Malvern) at 20 °C. Samples of STP10 wild type and mutants were prepared in an identical manner as described above. For Size-exclusion chromatography a buffer with 20 mM NaCitrate, 250 mM NaCl, 10% glycerol and 0.03% DDM adjusted to pH 5.5 was used. For the high pH data, STP10 C77A mutant was purified in an identical buffer using 20 mM MOPS and adjusted to pH 7.5. Fractions containing the purified protein were pooled and directly used for ITC experiments. Sample concentration was avoided to minimize any mismatch derived from empty detergent micelles. D-glucose was dissolved in the size-exclusion chromatography buffer and both protein and ligand were degassed prior to use. The sample cell was loaded with ~1800 μl of STP10 WT (50–80 μM) or STP10 C77A (20–40 μM) and titrated with a 5–10 fold higher concentration of D-glucose. A total of 36 injections of 8 μl aliquots were titrated into the protein sample. Each injection had a duration of 7 s and spaced with a 250 s interval. The stirring speed was set to 312 r.p.m. Data was corrected for nonspecific heat and analyzed using MicroCal Origin 7.0 software using a one-site binding model. The experiments were performed in triplicate and showed similar results.

**Yeast uptake assay**. For functional characterization, experiments were performed essentially as described by Sauer and Stadler[63]. In brief, the STP10 gene was subcloned into a p426MET25 vector[40] for constitutive expression and transformed into the S. cerevisiae hexose transport deficient strain, EBY-WV4000[64], using the lithium acetate/ PEG method. Transformed cells were plated in synthetic dropout media with 2% maltose and without uracil. Four to five colonies were used to inoculate 50 ml of synthetic dropout media with 2% maltose, without uracil and methionine and grown to an optical density at 600 nm (OD600) of ~1.5. Cells were washed twice with 25 mM NaPO$_4$ buffer pH 5.0, and resuspended in the same buffer to an OD600 of 10. The cells were dispensed into 1 ml aliquots, flash frozen and stored at −80 °C. For each reaction 20 μl of cell were mixed with 180 μl of 50 mM NaPO$_4$ adjusted to the pH intended for the experiment. For all assays pH was set to 5.0 unless otherwise stated. Cells were shaken in a thermomixer at 30 °C and tests were initiated by adding substrate. The reaction was stopped at given intervals by adding 700 μl ice cold water, and the reaction was filtered on mixed cellulose ester filters (0.8 μm pore size) and washed with an excess of ice water. Incorporation of radioactivity was determined by scintillation counting. For all assays 1 μCi [3 H]-D-glucose or 1 μCi [3 H]-2-deoxy-D-glucose (PerkinElmer, USA) was used a the radioactive tracer. Competition assays were performed with 10 μM D-glucose (or 10 μM 2-deoxy-glucose), pH dependent assays with 20 μM D-glucose and time-dependent uptake assays with 100 μM D-glucose. For competition assays all competing sugars were added in 25× excess (250 μM), and the inhibitors dissolved in DMSO and added as a 200× dilution at a final concentration of 500 μM (except CCCP 100 μM). For the determination of Km values, pH dependency, inhibition and substrate specificity, cells were incubated with [3 H]-D-glucose for 4 min to keep uptake in the linear range. For the Km value determination, the data was normalized to the predicted Vmax by fitting the data to Michaelis-Menten kinetics. The experiments were performed at least in triplicate and showed similar results. Data was analyzed with Graph Pad Prism 7.

**Yeast complementation assay**. Transformants were prepared as described above for the yeast uptake assay. Transformants were selected on SD (synthetic deficient) medium with 2% Maltose as carbon source and auxotrophic requirements. Cells were grown to OD600 of 0.6–1.0 in 5 ml of SD media without uracil and methionine supplemented with 2% Maltose. Cell suspensions were diluted to an OD600 of 0.5 and four five-fold serial dilutions were performed. Dilluted cell suspensions were plated in medium without uracil containing 0.02–2% of the desired carbon source and incubated at 30 °C for 5 days before photos were taken of the last three steps of the dilution series.

**Reporting summary**. Further information on experimental design is available in the Nature Research Reporting Summary linked to this article.

## Data availability:

Data supporting the findings of this manuscript are available from the corresponding author upon reasonable request. Coordinates and structure factors have been deposited in the Protein Data Bank with the accession number 6H7D.

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

## Acknowledgements

The authors acknowledge beamlines I24, I04-1 and I04 at the Diamond Light Source, where X-ray data were collected, as well as Max IV Laboratory, DESY-PETRA III and the Swiss Light Source for crystal screening. This work was supported by funding from the European Research Council (grant agreement No. 637372), the Danish Council for Independent Research (grant agreement No. DFF-4002-0052), the Carlsberg Foundation (CF17-0180), and an AIAS fellowship to B.P.P.

## Author contributions

P.A.P. did crystallization experiments and yeast uptake assays. T.F.C. did complementation assays, isothermal titration calorimetry and oocyte assays. B.P.P. supervised the project and processed crystallographic data. All authors contributed to analysis and paper.

## Additional information

**Competing interests:** The authors declare no competing interests.

