## [Peer Review File · Nature Communications]

Reviewers' Comments:

Reviewer #1:

Remarks to the Author:

This manuscript presents the structure of STP10, the first for any STP family of sugar transporters. STP transporters are wide-spread monosaccharide transporters in plants (with 14 members in Arabidopsis) and play important roles in many aspects of plant physiology. As importantly, STPs enable the storage of high amounts of sugar in fruits and are key players in host-pathogen interactions (for example defense against rust). As a result, understanding how STPs work at the molecular level will not only shine light on the sugar transport mechanism but also have important implications for biotechnological applications.

STPs have a very high affinity for monosaccharides and share low sequence similarity with GLUT sugar transporters. Paulsen et al. determined a high-resolution structure of STP10 (at 2.4 Å resolution) with the substrate glucose bound. Unexpectedly, the structure revealed a Lid domain that caps the extracellular surface and a disulfide bridge that locks the mobile transmembrane domains. In addition, these studies reveal a potential mechanism behind the high substrate binding affinity and the coupling between proton translocation and sugar transport. Structural studies are also nicely complemented by functional characterizations. In summary, the data are of high quality and well presented, and the findings are novel and exciting. This work will be of great interest to a broad audience interested in membrane biology, protein structure, or plant physiology.

There are however a few issues that need to be addressed. At a few places, the conclusions or claims need to be substantiated.

- 1) L43 is proposed to play a critical role in achieving high affinity to the substrate through hydrophobic interactions. This is plausible but still needs more experimental validation. (a) Does mutating an equivalent position in other related sugar transporters confer high substrate affinity? (b) Is the hydrophobic interaction important? For example, if the Leu is mutated to residues with similar size but differ in hydrophobicity, does it make a significant difference?
- 2) It is proposed that the cys mutation does not affect the substrate binding affinity but affects protonation. As the direct measurement of the binding affinity of cys mutant is only shown at one pH, the measurement should be done at different pH to demonstrate whether the effect of the mutation on K_m at different pH is linked to the K_d . In addition, there is no apparent entrance route for the substrate in the structure. With the disulfide bridge, it is not clear how the extracellular side can open up. Are there any alternative hypothesis?
- 3) The sugar transport is proton coupled. Is the transport rate linked to the proton gradient? This point was not clearly addressed in the manuscript. For example, in the uptake assay, at different pH, the proton concentration gradient is different too. It is likely that the transport rate is linked to both the proton concentration (pH) and its gradient across the membrane. This may impact the interpretations of several results (for example, Fig 3d and ED Fig 9a).
- 4) Competition assay is informative about the substrate specificity. But the direct measurement of the uptake for each putative substrate needs to be shown. It is not uncommon that molecules could compete with the substrate but can not be transported. In the yeast complementation assay, it is not clear why mannose and glucose behave quite differently on the mutant. For example, for cys mutants, yeast grows even better than the WT in the presence of mannose but not glucose.

Reviewer #3:

Remarks to the Author:

Summary:

Herein the authors have expressed, purified and crystallized one of the plant Sugar Transport Proteins (STPs) – STP10. The X-ray crystal structure of this sugar/H⁺ symporter was solved to 2.4 Å resolution and represents a closed occluded conformation of the protein. The structure reveals a bound glucose molecule at the interface between two domains which are locked together with a disulfide-bonded LID domain. A handful of functional assays were presented, including competition assay in yeast with radioactively labeled glucose and an assay in *Xenopus* oocytes. These assays allowed for the analysis of substrate affinity (K_m and K_d values) as well as for investigating substrate specificity in different STP10 mutants and testing of transport inhibitors. The authors have proposed a mechanism of glucose coupling to protonation of the identified site. This is the first reported plant STP structure and novelty of this protein resides in the unusual lid domain that sits on top of the substrate binding site.

Minor concerns:

- The manuscript title is too generic. Crystal structure and mechanism of plant Sugar Transport Proteins .. Perhaps "The Crystal structure of STP10 informs on the mechanism of plant Sugar Transport Proteins"
- When describing the differences between STP10 and other sugar transporters, the lid domain becomes an obvious difference. As in Extended figure 7.. it would be important to show a structural alignment with these other transporters to highlight the lid domain –if it is indeed completely novel in this class of proteins.
- Line 92-95: At the first mention of K_m and K_d values, add 1 sentence of why these values are similar in this transporter case (since those report on 2 different constants and would be different in enzymes for examples).
- Line 95-96 "Competition assays confirm galactose and mannose as substrates as well (Fig. 2d)".: Mention that it is an assay where readout is radiolabeled glucose. Mention that those sugars are transported in addition to glucose. Most importantly though: explain why mannose (75%) is considered a substrate and xylose and fructose (80-85%) are not?
- Line 96-97 "Growth assays show transport of both mannose and fructose at high sugar concentrations (Extended Data Fig. 6b).": Mention that it also shows glucose and name it properly as growth complementation assay. Fructose does not look better than glucose at 0.2% and only minimally at 2%. This is highly inconclusive. Explain or attempt a different assay.
- Line 100 "extended Fig 6c and 6d": Explain why some inhibitors work and some don't.
- Line 100 "STPs display very high affinity in the μM range for their substrates.": Where are the data to support this? You only show K_m and K_d for glucose.
- Line 103-105 "This tight interaction surface is not found in human sugar facilitators or bacterial sugar/H⁺ symporters^{14–16} (Extended Data Fig. 7). Using hydrophobic interactions to boost affinity is a common theme for high affinity protein-ligand and protein-protein interactions".: This figure shows hydrophobic pockets for other transporters as well. Why do you claim your site is unique? Is it only the closer distance of L43 to glucose in your structure? Elaborate or remove this sentence.
- Line 110 "These are the sole candidates for the proton donor/acceptor pair needed for proton translocation": Reference your Ext. Data Fig. 8. and mention that those are the only possible charged residues. Given your bold statement, propose/suggest how H⁺ translocation would work (symport with glucose).
- Line 112 "...abolishes transport": Not true: R142K does not do that. Also, fructose transport looks fine at 0.02% and 0.2%. Edit. At this point it looks like
- The yeast growth complementation assay yields inconclusive results, that are often not reported accurately by the authors. The interpretation needs to align with the results or the assay needs to be removed or redesigned.
- Line 134-135 "At high substrate concentration cell viability was unchanged (Extended Data Fig. 6b).": It was changed – for higher and high mannose and for 2% fructose.
- Line 152 "...C domain interactions mediate specificity". Why and how?
- Line 153 "...hydrophobic interactions and the LID domain explain...": be specific about the

interactions. This whole sentence does not read well.

- Line 164: Why does this structure explain high specificity? What experiments do you have to prove this? You showed transport of multiple sugars thus far...

Other Minor Suggestions:

- Draw lines to indicate membrane around your figures with protein structure on both sides of the protein, use thicker lines.
- Figure 2: Panel A could go to Figure 1. Why are there no error bars on panel C?
- Figure 4: Mark Phe39 on your figures. The first sentence of the legend is already in the main text.
- Extended Fig 1 and 2. Red and pink labels are too close. Change colors.
- Extended Data Fig. 5. Could be added to figure 1 in the main text.
- Extended Data Fig. 6. Move this info to the main text: "Transport is dependent on the proton gradient as shown by the use of the proton gradient decoupler CCCP, the protein can transport the non-metabolized glucose analog 2-deoxyglucose, and the protein is sensitive to some of the inhibitors known from the field of bacterial and human sugar transport."
- Extended Data Fig. 9. Panel A should replace Fig 3d in the main text. No error bars for Panel C. As should the statements in the figure legend to the discussion in the maintext: "Increasing Lid domain flexibility by removing the disulfide bridge is linked to protonation and does not directly change affinity towards glucose (as seen for low pH). However, at higher pH, Asp42 does not become protonated easily, leading to lower turnover and lower glucose affinity."
- Line 41 "apoplastic sugar depletion": add 1 more sentence to clarify the phenomena.
- Line 47-51: the intro would read better if this section came after "...throughout the plant kingdom" in line 45.
- Line 54 "...pH is alkalized": add 1 more sentence on what this alkalization accomplishes.
- Line 58: In addition to Ext. Fig. 2, prepare an Ext. Table. 2 with % identity table (similar to Ext. Table 1)
- Line 60-63: This should be in the introduction.
- Line 88 "...C6...": mention that this is the carbon coordinate of glucose; apply to other places in the text where only C4, C5 etc, is used.
- Line 98: Reference figure 2D at the end of the sentence in this line.
- Line 115-116 "Interestingly, rust resistance in wheat can be conferred by the Lr67 gene (encoding an STP) which contains a mutation at exactly the equivalent of the Arg142 position".: So a positive trait emerges as a result of a mutation that should abolish proper STP function? Explain.
- Line 120 "This is consistent with the low pH of the crystallization...": Pka of side chain Asp is 3.9 and your buffer 4.5. You need to make a point about the local amino acid profile and its influence on pKas of amino acids such as your Asp, to convince readers that instead of being mainly deprotonated, your Asp is protonated.
- Line 130 "(Fig. 2a and Fig. 3a).": These features (Phe 79...)are not visible on these figures, use Fig. 3b instead (Add trp 202 which is missing). Or make a new figure based on 3b.
- Line 132. From your figures, it looks like Asp42 could be accessed by water. Explain.
- Line 138: "...confers pathogen resistance...": Why does the mutation help?
- Line 146: Described rocker-switch model if you want to challenge it (briefly).
- Line 149: Give pH range for the transport.
- Line 169-170: How do your experiments 'shed new light on sugar transport'?

Reviewer #4:

Remarks to the Author:

The manuscript by Paulsen et al. describes the first structure of a member of the important MST sugar transporter family from plants named STP10 at a resolution of 2.4 Å. Given the importance of STPs and the current discourse regarding their role in pathogen susceptibility this work is of high relevance and lays the basis for a better understanding of the currently difficult to understand

phenomena described for example by Moore et al or Yamada et al. Moreover, the transporters in this family play crucial roles in many aspects of plant physiology, equal in importance as human GLUTs. The gained insights into the STP structure are also relevant in the context of understanding how some of these transporters function as uniporters (GLUTs), others as proton symporters (STPs). STPs are phylogenetically related to the human GLUTs and the structure solved here now enables us to compare structures, functions and regulation. Exciting new insights include the identification of a putative binding site for the substrate, the presence of an unusual lid with a disulfide bridge, and the conserved presence of an intracellular ICH domain. In addition the authors test the role of the two lid Cys for activity. They observe that mutation of cys reduces activity. They predict residues that may be involved directly in proton coupling and show that mutation reduces activity. They find the protein in an outward facing occluded state and generate a model for the transport mechanism, however this model needs to be interpreted with maximal caution in the light of the unusual lid domain.

The reviewer is not a structural biologist, thus can not judge the quality of the structure, however it appears important to carefully check the LID domain, since it is a unique feature. Also it should be noted that it is conceivable that the disulfide bridge is not always present in vivo.

There is quite some literature that could be integrated here to discuss the structure and the residues relevant for activity. Sauer had altered the substrate specificity of an STP. Tanner had characterized residues important for substrate recognition in the Chlorella homolog, which is called HUP1. An important manuscript to be mentioned in the context of the ICH domain is the loss of function caused by deletion of the highly conserved C-terminus (conservation of cytosolic C termini is an exception, AMTs have highly conserved C termini important for function, STPs apparently as well). Examples are Grassl et al. 2000, Will et al. 1998, Buettner Sauer 2000. Very interesting is for example the finding from Tanner that deletion of part of the C-terminus in HUP1, an ancestor of the STPs leads to loss of activity, a mutation that also leads to loss of resistance in STP13 (Moore). This should all be discussed in the light of the possible role of the ICH domain.

Major comments:

1. The authors fail to cite key references regarding the role of STPs (Norholm 2006 role in cell death; Schofield 2009 nitrogen use efficiency and growth, Yamada et al 2016 Science STP1 and 13 and Pseudomonas, signaling, interaction with FLS2 and BAK1; Lemmonier 2014 Botrytis resistance which is exact opposite of what Moore finds in wheat). They also do not discuss homo-oligomerization at all, which Moore claims to be key to resistance in the dominant resistance seen in hexaploid wheat. It is interesting that also other structural analyses of for example GLUTs have avoided to discuss oligomerization – e.g. Carruthers had suggested that GLUTs function as dimers of dimers, in which two subunits always switch together in the antiparallel orientation, respectively. It appears that none of the structures here shows signs of di- or higher oligomeric states. Important to mention.
2. The authors do not compare structures to GLUTs (R0 etc) and do not highlight possible similarity and possible major structural differences. At least in the supplement a figure that focuses on the binding site should be shown side by side with binding sites from other members of MFS sugar transporters. They should discuss binding site interactions in more detail (taking also Sauer and Tanner data into account).
3. There is no experimental validation of the structure by testing the role of proposed substrate binding sites or other important domains. This is very easy in this case by using at least qualitative data from yeast complementation.
4. There is also a major error in the interpretation regarding the residues that are important for dominant rust resistance (LR67) in STP13. I admit I also had to read the paper multiple times and look at their background papers before this became fully clear. Note that Moore uses LR67 for both the susceptible and the resistant version and marks them res variants. Moore et al. describe that

Gly144 and Val387 are key to resistance in wheat (although from their data still unclear whether both are required). Then they describe suppressor mutations that lead back to susceptibility! These include 8 mutations listed in the Moore paper. They state that Gly144 in the susceptible version (the quasi wild type) is conserved in all STP13s. The resistant version contains an Arg in that position, creating a double R at the position in the membrane with possibly dramatic effects on the transmembrane helix. Thus the Arg142 discussed for STP10 in this manuscript is right next to the equivalent of the Gly144 Arg described by Moore. Similarly, the statement that the equivalent of Cys77 in wheat STP13 is relevant to resistance is not correct. The suppressor screen identified mutations that suppress resistance, and the most likely explanation is that all suppressor mutations lead to a loss of function, retaining the protein but eliminating the activity, which leads to resistance. This needs to be clarified and corrected throughout the text and discussed in more detail.

5. The predicted lid and the disulfide bridge in the predicted lid could potentially be very exciting. The authors tested the importance of the Cys residues for activity. They observe a reduction in activity, but while interesting, these experiments do not address the key question – what is the lid's role and is it regulated. They rather make strong statements that are not warranted by any data, for example that the finding of the lid challenges the rocker switch model. It is important to test redox dependence of the activity, as shown to be important for other transporters, for example the regulation of the yeast calcium channel by redox regulation (Cch1p glutathionylation). Is it not likely that this disulfide bridge is biologically and structurally relevant. Could it be that this helped to lock the structure into the observed conformation and occurred during purification? Can it be reduced in the structure?

Minor comments

1. The title is not informative and unintentionally misleading. It is not the authors fault, since STP stands for Sugar Transport Protein, but for the broad audience misleading and unclear. Important to change the title to make clear it's a plant STP or MST related to GLUTs.
2. Summary: the study does not identify the proton donor acceptor pair, it provides circumstantial evidence that these might be the residues. Needs to be tuned down
3. STPs are not active, they are secondary active.
4. The statement that STP10 has a very high selectivity for three substrates is odd, and not warranted by the data, since only a limited number of soluble sugars and glycosides was tested. SUTs have been found to transport for example esculin, fraxin and helicin.
5. High glucose concentrations (high is relative) do not have these effects in all experiments and plants. And the sentence that states this is odd since it states sugars stunt plants as do STPs...STPs are just transporters. Do you mean that the plants become more sensitive to glucose in particular when STPX is overexpressed?
6. This effect is confirmed – reference is unclear and confirmed is too strong, replace by supported
7. The statement that the large tolerance for pH created by the lid is directly related to STPs physiological function is too strong and unclear, different members have different functions, and the pH profile of measured activity is created by a combination of the pH profile of the protein itself and the proton coupling (and if tested in yeast influenced by the acidification by the yeast when it receives sugars).

Reviewers' comments

(reply from authors in blue.)

Reviewer #1 (Remarks to the Author):

This manuscript presents the structure of STP10, the first for any STP family of sugar transporters. STP transporters are wide-spread monosaccharide transporters in plants (with 14 members in Arabidopsis) and play important roles in many aspects of plant physiology. As importantly, STPs enable the storage of high amounts of sugar in fruits and are key players in host-pathogen interactions (for example defense against rust). As a result, understanding how STPs work at the molecular level will not only shine light on the sugar transport mechanism but also have important implications for biotechnological applications.

STPs have a very high affinity for monosaccharides and share low sequence similarity with GLUT sugar transporters. Paulsen et al. determined a high-resolution structure of STP10 (at 2.4 Å resolution) with the substrate glucose bound. Unexpectedly, the structure revealed a Lid domain that caps the extracellular surface and a disulfide bridge that locks the mobile transmembrane domains. In addition, these studies reveal a potential mechanism behind the high substrate binding affinity and the coupling between proton translocation and sugar transport. Structural studies are also nicely complemented by functional characterizations. In summary, the data are of high quality and well presented, and the findings are novel and exciting. This work will be of great interest to a broad audience interested in membrane biology, protein structure, or plant physiology.

We thank the reviewer for the enthusiasm for the presented work.

There are however a few issues that need to be addressed. At a few places, the conclusions or claims need to be substantiated.

1) L43 is proposed to play a critical role in achieving high affinity to the substrate through hydrophobic interactions. This is plausible but still needs more experimental validation. (a) Does mutating an equivalent position in other related sugar transporters confer high substrate affinity?

We agree with the reviewer that that is a very promising point-of-attack for this observation. It is beyond the scope of the current paper to identify other sugar transporters with good expression for yeast uptake assays, establish assay protocols and do the mutations. We have been looking into the literature and (with the help from reviewer #4's comments) identified a paper from the group of Tanner (Will et al JBC 273(1998)) to support our claim. This is now included in the manuscript (line 163):

“Supporting this key role of Leu43, a similar pattern is observed in related sugar transporters HUP1 and HUP2 from the algae Parachlorella kessleri, where mutating this position changes affinity 20-fold depending on side chain hydrophobicity (Extended Data Fig 1)³⁹. This M1b-linked mechanism to control affinity thus appear to be conserved not only in STPs but also in closely related protein families outside the plant kingdom (Extended Data Fig. 1).”

(b) Is the hydrophobic interaction important? For example, if the Leu is mutated to residues with similar size but differ in hydrophobicity, does it make a significant difference?

This is a very interesting point. To address this we have made new mutant where we have replaced the Leu43 with asparagine. Leu and Asn have the same number of atoms in the same bond position, but Asn have a polar nature compared to the hydrophobic nature of the Leu side chain moiety. Using this mutant (L43N), we

have done the uptake assay and calculate a K_m of $\sim 149 \mu M$. This is half of the L43A mutant ($391 \mu M$, fig 3c). Based on this we tentatively conclude that the hydrophobic nature of the side chain is important as originally stated contributing to about half of the affinity, compared to a complete removal of the interaction. This assay have been included as Extended Data Figure 9a, and the manuscript have been modified to make this point on line 158:

“This mechanism is supported by the L43A mutant, which dramatically reduces STP10 affinity for glucose and turns STP10 into a low affinity transporter (K_m 391 μM) (Fig. 3c). A corresponding mutant L43N which replaces the hydrophobic interaction surface to a polar one of similar size, also leads to a significant decrease (K_m 149 μM), highlighting that the hydrophobic aspect of the interaction is a key contributor to affinity (Extended Data Fig. 9a).”

2) It is proposed that the cys mutation does not affect the substrate binding affinity but affects protonation. As the direct measurement of the binding affinity of cys mutant is only shown at one pH, the measurement should be done at different pH to demonstrate whether the effect of the mutation on K_m at different pH is linked to the K_d .

This is a good point, and we have done an experiment to look further into this aspect. We have done ITC on the C77A mutant at pH 7.5 which gives no significant change in K_d . This demonstrates that the change in K_m caused by pH shift is not apparently linked to the K_d . However as ITC probes both inward and outward conformations in the experimental setup, those results should be interpreted with some care. We have added the experiment as an extra panel to Extended Data Figure 9d, and added the following to the main text (line 186):

“By breaking the disulfide bridge and increasing flexibility, the proton/donor acceptor pair becomes much more sensitive to the extracellular pH, either directly through a change in Asp42 pK_a value or through a requirement for a stronger proton gradient to drive transport, and this indirectly affect substrate turnover at higher pH, without affecting K_d (Extended Data Fig. 9).”

In addition, there is no apparent entrance route for the substrate in the structure. With the disulfide bridge, it is not clear how the extracellular side can open up. Are there any alternative hypothesis?

It is correct that the classical entry pathway found in eg. GLUT3 and other sugar transporters is not readily visible. We believe that this is also part of the impact of our findings. Our current proposal is that the substrate can enter by rearrangements of the helices and the lid, that we do not observe in this structure. However it is clear that small rearrangements of especial the loop section of the helix-helix-loop-helix structure of the lid domain could lead to a (narrow) opening that would allow for sugar to enter the site where the current cavity entrance (Fig 3a) is visible. However a complete splayed out outward open form would be impossible, but is not necessary for substrate to enter (this type of open form, while observed in a few MFS structures (eg. LacY and GlpT) has never been demonstrated to be necessary for transport). New structures of an outward empty form might help to examine this point in the future.

We have added the following to the text (line 194) to clarify this point:

“Rearrangements of the N domain and the Lid domain must occur to allow the monosaccharide substrate to bind, but there is no clear entry pathway as seen in other Major Facilitators¹⁸. However, smaller rearrangements of M1, M5, M8 and the loop region of the lid domain could create an entry pathway to the central binding site (Fig. 3a and Extended Data Fig. 7).”

3) The sugar transport is proton coupled. Is the transport rate linked to the proton gradient? This point was not clearly addressed in the manuscript. For example, in the uptake assay, at different pH, the proton concentration gradient is different too. It is likely that the transport rate is linked to both the proton

concentration (pH) and its gradient across the membrane. This may impact the interpretations of several results (for example, Fig 3d and ED Fig 9a).

This is an excellent point, and we have already spent significant time pondering how to address this. Since our uptake assays are made in oocytes or yeast, it is currently impossible for us to manipulate the internal pH, and thus deconvolute the gradient from the actual proton concentration. Thus the changes we observe in transport could be caused by either pKa changes (i.e. proton concentration effect) or by the actual proton motive force available to drive transport. We plan to make future experiments to deconvolute this aspect of STP10 transport.

This is addressed in the following text of the manuscript (line 186):

“By breaking the disulfide bridge and increasing flexibility, the proton/donor acceptor pair becomes much more sensitive to the extracellular pH, either directly through a change in Asp42 pKa value or through a requirement for a stronger proton gradient to drive transport, and this indirectly affect substrate turnover at higher pH, without affecting Kd (Extended Data Fig. 9).”

4) Competition assay is informative about the substrate specificity. But the direct measurement of the uptake for each putative substrate needs to be shown. It is not uncommon that molecules could compete with the substrate but can not be transported.

While we agree that measuring actual transport with different substrate would be optimal, we (due to availability of radio-labeled sugars in Denmark), cannot currently perform these experiments. Negotiations are ongoing to obtain these compounds but could take significant time. We agree that the competition assay shows only competition and not actual transport. However the complementation assay does demonstrate transport, and taken together these two experiments, while not perfect, indicate that our proposed substrates are transported and not inhibitors. These suggested substrates (glucose, mannose and galactose) are consistent with a report for STP10 substrates in the literature (Rottmann et al J. exp. botany 67 (2016)). Surprisingly the complementation assay seems to indicate fructose transport as well but only at extremely high sugar concentrations. See also our comments to reviewer #3 on this topic.

We have adjusted the text to tone down actual transport (line 109).

“Competition assays using radioactively labeled glucose confirm galactose and mannose as potential substrates (Fig. 2d)”

In the yeast complementation assay, it is not clear why mannose and glucose behave quite differently on the mutant. For example, for cys mutants, yeast grows even better than the WT in the presence of mannose but not glucose.

See also our comments to reviewer #3. This is an interesting point. First of, the level of growth of the cys mutants must be considered equivalent to the growth level in WT in mannose (extended data fig. 6b). The complementation assay is not sensitive enough to clearly distinguish between growths levels at that resolution. However it is absolutely correct that growth in high mannose is much more pronounced than with the same levels of glucose and this has also puzzled us for some time. Inhibition of glucose transport at high glucose concentrations have been reported for STPs previously (eg. Scholz-Starke et al Plant Phys 131 (2003); and very clearly in Yamada et al JBC 286 (2011)), while other sugars do not appear to have a similar effect in complementation assays. It is unclear what causes this effect. As this paper's focus is elsewhere, we have not lingered on this finding in the manuscript, but only state (line 112):

“High glucose concentrations (>10 mM) appear to inhibit yeast growth as also reported previously for other STPs in growth complementation assays^{27,31}.”

Reviewer #3 (Remarks to the Author):

Summary:

Herein the authors have expressed, purified and crystallized one of the plant Sugar Transport Proteins (STPs) – STP10. The X-ray crystal structure of this sugar/H⁺ symporter was solved to 2.4 Å resolution and represents a closed occluded conformation of the protein. The structure reveals a bound glucose molecule at the interface between two domains which are locked together with a disulfide-bonded LID domain. A handful of functional assays were presented, including competition assay in yeast with radioactively labeled glucose and an assay in *Xenopus* oocytes. These assays allowed for the analysis of substrate affinity (K_m and K_d values) as well as for investigating substrate specificity in different STP10 mutants and testing of transport inhibitors. The authors have proposed a mechanism of glucose coupling to protonation of the identified site.

This is the first reported plant STP structure and novelty of this protein resides in the unusual lid domain that sits on top of the substrate binding site.

We thank the reviewer for the support of our work.

Minor concerns:

- The manuscript title is too generic. Crystal structure and mechanism of plant Sugar Transport Proteins .. Perhaps “The Crystal structure of STP10 informs on the mechanism of plant Sugar Transport Proteins”

In hindsight we agree the initial title was too generic. We have changed the title to:

“Crystal structure of the plant proton/sugar symporter STP10 illuminates the mechanism behind sugar uptake in the Monosaccharide Transporter Superfamily”

- When describing the differences between STP10 and other sugar transporters, the lid domain becomes an obvious difference. As in Extended figure 7.. it would be important to show a structural alignment with these other transporters to highlight the lid domain –if it is indeed completely novel in this class of proteins.

It is indeed novel. We have now expanded Extended Data Figure 7 to show the overall fold of Xyle, GLUT3 and STP10, highlighting the Lid domain region. This panel has been aligned on the N domain since there is no Lid domain to align by in Xyle or GLUT3.

- Line 92-95: At the first mention of K_m and K_d values, add 1 sentence of why these values are similar in this transporter case (since those report on 2 different constants and would be different in enzymes for examples).

It is correct that in enzymes these two values are expected to be different (except for the somewhat special case where $k_{-1} \gg k_2$). For transporters the observation of $K_m \approx K_d$ is relatively normal, as also exemplified for the xylose transporter Xyle in Deng et al Nature 490 (2012)(supp. fig 3a+c). The kinetics of transport are rather complicated, and have been discussed several places in detail (eg. Vivian & Polli Eur, J Pharm Sci. 64, (2014) for a recent attempt).

Results in the field are further complicated by the matter that it is customary to use ITC or similar techniques to obtain K_d values. These techniques probe both inward and outward conformations in almost all experimental setups (the tested sample will be a mixture of inward and outward facing protein). Thus K_d results in the transporter field (including ours) should be interpreted with some care. We have extended the sentence to highlight this (line 107).

“The apparent K_d of glucose binding to STP10, as determined by isothermal titration calorimetry on the purified sample, is in the same range as the K_m , as has been observed previously for other sugar transporters¹⁹ (Fig. 2c).”

- Line 95-96 “Competition assays confirm galactose and mannose as substrates as well (Fig. 2d)”.: Mention that it is an assay where readout is radiolabeled glucose. Mention that those sugars are transported in addition to glucose.

We have changed the sentence to (cf also reviewer #1 comment 4) (line 109):

“Competition assays using radioactively labeled glucose confirm galactose and mannose as potential substrates (Fig. 2d).”

Most importantly though: explain why mannose (75%) is considered a substrate and xylose and fructose (80-85%) are not?

Mannose was in the original manuscript presented as a substrate because 1) the difference to WT in the competition assay was statistically significant (just barely, $p = 0.02$) while xylose and fructose were not (fig. 2d), 2) Mannose as a substrate is observed by the growth complementation assays (see point below). 3) Mannose as a substrate is already reported in the literature (Rottmann et al J. exp. botany 67 (2016)). However we do agree with both reviewer #1 and reviewer #3 that full evidence for transport would require the use of radioactively labeled mannose or similar. We have rephrased the text to suggest mannose as a *potential* substrate as mentioned above, and we have added the p values to fig 2d to help the reader evaluate the strength of the evidence that mannose inhibits glucose uptake.

- Line 96-97 “Growth assays show transport of both mannose and fructose at high sugar concentrations (Extended Data Fig. 6b).”: Mention that it also shows glucose and name it properly as growth complementation assay.

The assay show no or very little transport of Glucose at high concentration (>10 mM). We have adjusted the text to clarify our point. We have also added the concentration in mM to Extended Data Fig 6b, to better highlight the point that a growth complementation assay is done at significantly higher substrate concentration than the K_m of the transporter in question. This type of assay is relatively normal in the field (e.g. Yamada et al JBC 286 (2011)). The high sugar concentration appear to have deleterious effects with glucose, but not for 'suboptimal' substrates like mannose and apparently also fructose. However a high concentration of the carbon-source is needed to allow the cells to grow to a point where the colonies are visible.

The text now reads (line 110)

“Furthermore, growth complementation assays show uptake of glucose at lower sugar concentration (~1 mM), and for both mannose and fructose at higher sugar concentrations (>10 mM) (Extended Data Fig. 6b). High glucose concentrations (>10 mM) appear to inhibit yeast growth as also reported previously for other STPs in growth complementation assays^{27,31}. “

Fructose does not look better than glucose at 0.2% and only minimally at 2%. This is highly inconclusive. Explain or attempt a different assay.

We agree that we do not see any significant transport of fructose at 0.2% concentration. However at 2% concentration of fructose the STP-WT clearly survives while the control does not. In this respect 2% fructose allows survival while 2% glucose inhibits survival. Likely fructose is a very low affinity substrate for STP10,

only being transported at extremely high concentrations. Inhibition of yeast growth by high glucose concentration is a peculiar feature of STPs that have been reported several times before (eg. Scholz-Starke et al Plant Phys 131 (2003); and very clearly in Yamada et al JBC 286 (2011)), but the cause is still unknown, and beyond the scope of this paper.

- Line 100 “extended Fig 6c and 6d”: Explain why some inhibitors work and some don't.

The tested inhibitors are known from the literature on bacterial and human sugar transporters, and have not been tested for plant sugar transporters before to our knowledge. There are significant sequence differences between plant and human/bacterial transporters, so it is not surprising that not all the inhibitors will have the same effect.

We have now expanded the sentence to read (line 121):

*“Transport is dependent on the proton gradient as demonstrated by the use of the proton gradient decoupler Carbonyl cyanide *m*-chlorophenyl hydrazone (CCCP). The protein can transport the non-metabolized glucose analog 2-deoxyglucose, and appear to be sensitive to only some of the inhibitors known from bacterial and human sugar transport (Extended Data Fig. 6e and 6f). ”*

- Line 100 “STPs display very high affinity in the μM range for their substrates.”: Where are the data to support this? You only show K_m and K_d for glucose.

The reviewer is correct that this sentence was too broad in scope. We have modified it to the following (line 127):

“The μM affinity of STP10 for glucose can be explained by ... ”

- Line 103-105 “This tight interaction surface is not found in human sugar facilitators or bacterial sugar/ H^+ symporters^{14–16} (Extended Data Fig. 7). Using hydrophobic interactions to boost affinity is a common theme for high affinity protein-ligand and protein-protein interactions”.: This figure shows hydrophobic pockets for other transporters as well. Why do you claim your site is unique? Is it only the closer distance of L43 to glucose in your structure? Elaborate or remove this sentence.

It is the combination of the significantly closer distance and well as the fact that we have hydrophobic leucine in the position while in both XylE and GLUT3 (only two other structures known in an outward facing conformation) the corresponding residue is a polar threonine.

We have modified the text to better explain our point (line 129):

“This tight and hydrophobic interaction surface is not found in human sugar facilitators or bacterial sugar/ H^+ symporters, where the interaction distance is longer and the corresponding residue is polar^{17–19} (Extended Data Fig. 7). Using tight hydrophobic interactions to boost affinity is a common theme for high affinity protein-ligand and protein-protein complexes^{33,34}. ”

We now further demonstrate that the hydrophobic nature of this interaction forms a significant part of the affinity with a new L43N mutant. Leu and Asn have the same number of atoms with the same bond architecture, but Asn have a polar nature compared to the hydrophobic nature of the Leu side chain moiety. This assay have been included as Extended Data Figure 9a, and the manuscript have been modified to make this point on line 158:

“This mechanism is supported by the L43A mutant, which dramatically reduces STP10 affinity for glucose and turns STP10 into a low affinity transporter (K_m 391 μM) (Fig. 3c). A corresponding mutant L43N which replaces the hydrophobic interaction surface to a polar one of similar size, also leads to a significant

decrease (K_m 149 μM), highlighting that the hydrophobic aspect of the interaction is a key contributor to affinity (Extended Data Fig. 9a). ”

- Line 110 “These are the sole candidates for the proton donor/acceptor pair needed for proton translocation”: Reference your Ext. Data Fig. 8. and mention that those are the only possible charged residues.

This is a key observation, and we mention it in the previous sentence. The text read (line 136) *“Here we find the only two buried charged residues in the transmembrane region, Asp42(M1b) and Arg142(M4) (Fig. 1a and 1c, and Extended Data Fig. 8). These are the sole candidates for the proton donor/acceptor pair needed for proton translocation³⁵. “*

Given your bold statement, propose/suggest how H⁺ translocation would work (symport with glucose).

We propose our model for how proton/sugar translocation would occur in the end of the manuscript. Our data suggest that the Asp becomes protonated and this allows for the movement of the M1b helix towards the substrate binding site, thus leading to tight coupling between proton and sugar binding. This conformation has a lower energy barrier to transverse into the inward facing conformation, where release of the proton (due to the proton gradient) will release the sugar as the M1b helix returns to its original position driven by the interplay between asp42 and arg142 (the proton donor/acceptor pair). The concept of a proton donor/acceptor pair in the context of proton transmembrane transport is reviewed in Buch-Pedersen et al *Pflüg. Arch. Eur. J. Physiol.* **457** (2009).

The identification of a proton donor/acceptor pair in M1/M4 (or M7/M10 which is topologically identical due to the pseudo-twofold of the MFS) is a general observation in several families of major facilitators, including XylE, GlcPse, as well as the lactose transporter LacY and phosphate and nitrate transporters (find references to all these in the manuscript if desired).

We state this in line 141:

“ The position is similar to the position of proton donor/acceptor pairs in other Major Facilitator proton driven symporters like the bacterial xylose/H⁺ symporter XylE³⁶, and the glucose/H⁺ symporter GlcPse¹⁷ (~27% sequence ID to STP10). ”

- Line 112 “...abolishes transport”: Not true: R142K does not do that.

We thank the reviewer for helping us clarify this important point. If removing either of the two charged residues (mutation to Ala) there is no transport. It is correct that R142K does not abolish transport (but only reduces it to a very low level), while D42E does. We believe that this shows that while the proton binding site asp42 is very sensitive to the size of the side chain, the positively charged residue is more flexible. The key is to have a positive charge buried here that can help modulate the pKa value of Asp42, not the exact nature of the side chain.

We have expanded this sentence to better explain our point (line 139):

“This key role is supported by mutating either Asp42 or Arg142 to alanine which abolishes transport. Arg142 seems to be somewhat more resilient to change as a mutation to lysine does still allow for minimal transport (Extended Data Fig. 6b). ”

Also, fructose transport looks fine at 0.02% and 0.2%.

We respectfully disagree with this. The fructose transport for both WT and the mutants at 0.02% and 0.2% is almost identical to the negative control. Only at high (2%) fructose concentrations is there a discernible

difference between the negative control and e.g. STP10 (wt). We have repeated this experiment several times, and we observe the same pattern every time.

Edit. At this point it looks like

- The yeast growth complementation assay yields inconclusive results, that are often not reported accurately by the authors. The interpretation needs to align with the results or the assay needs to be removed or redesigned.

We agree that there were shortcomings in the earlier very brief summary of the experiment. We hope the reviewer will agree that the new text does better justice to the experimental results.

- Line 134-135 “At high substrate concentration cell viability was unchanged (Extended Data Fig. 6b).” It was changed – for higher and high mannose and for 2% fructose.

We agree our previous statement was not precise enough. However within the noise of the complementation assay there is no change between the growth of WT and the Cys mutants in either concentration of mannose and if any in 2% fructose, only very slightly. We have adjusted the sentence to better clarify and tone down our statement (line 175):

”At the high substrate concentrations used in the growth complementation assay cell viability appear virtually unchanged in all settings (Extended Data Fig. 6b). ”

- Line 152 “...C domain interactions mediate specificity”. Why and how?

This statement is based on the structural observations. Why and how is explained in detail earlier in the manuscript in line 98f. We have also adjusted the text to better explain our point at the end (line 202):

“... the polar C domain interactions recognize the specific hydroxyl groups of the substrate and thus can mediate specificity. ”

- Line 153 “...hydrophobic interactions and the LID domain explain...”: be specific about the interactions. This whole sentence does not read well.

We agree this sentence was a bit clunky. We have deleted it as we find it was redundant. The same point is made below in the conclusion.

- Line 164: Why does this structure explain high specificity? What experiments do you have to prove this? You showed transport of multiple sugars thus far...

We agree that this statement was overstated and we have toned it down. It now reads (line 212):

“In particular the structure explains high sugar affinity and identifies and couples the proton-motive force to sugar transport. ”

Other Minor Suggestions:

- Draw lines to indicate membrane around your figures with protein structure on both sides of the protein, use thicker lines.

We have drawn lines on all structural figures to help orient the reader.

- Figure 2: Panel A could go to Figure 1. Why are there no error bars on panel C?

We would prefer to keep Extended Data Fig. 5 and Fig. 2b apart from Fig. 1 to give this figure a cleaner look.

There are no error bars on the ITC data because it is not methodologically possible to combine data-points from several ITC runs. We did reproduce the ITC experiments 3+ times and got very similar results every time. It is, to our understanding, normal procedure for ITC experiments to report a representative trace without error bars.

For a few examples see the following refs:

doi: 10.1074/jbc.M610075200 (Arginine-Agmatine Exchange Transporter)

doi: 10.3109/09687688.2012.696733 (glycine receptor)

doi: 10.1038/nature11683 (TatC core of the twin-arginine protein transport system)

doi: 10.1074/jbc.M112.391128 (NhaA Na⁺/H⁺ Antiporter)

doi: 10.1074/jbc.RA118.001796 (Salmonella effector SseK3)

doi: 10.1016/j.celrep.2017.10.053 (SXN27-PTEN complex)

doi: 10.1038/nature11524. (XylE glucose transporter (supp material))

- Figure 4: Mark Phe39 on your figures. The first sentence of the legend is already in the main text.

We have marked Phe39 and removed the first sentence.

- Extended Fig 1 and 2. Red and pink labels are too close. Change colors.

We have changed the pink label to light blue.

- Extended Data Fig. 5. Could be added to figure 1 in the main text.

We would prefer to keep extended data fig. 5 and fig. 2b away from fig. 1 to give this figure a cleaner look. While extended data fig. 5 is relevant for the structural explanation of STP10, it is not a key figure to understand the main insights presented that relate to the Lid domain and substrate affinity.

- Extended Data Fig. 6. Move this info to the main text: "Transport is dependent on the proton gradient as shown by the use of the proton gradient decoupler CCCP, the protein can transport the non-metabolized glucose analog 2-deoxyglucose, and the protein is sensitive to some of the inhibitors known from the field of bacterial and human sugar transport."

We have moved the information as suggested. The main text now reads (line 121):

"Transport is dependent on the proton gradient as demonstrated by the use of the proton gradient decoupler Carbonyl cyanide m-chlorophenyl hydrazone (CCCP). The protein can transport the non-metabolized glucose analog 2-deoxyglucose, and appear to be sensitive to only some of the inhibitors known from bacterial and human sugar transport (Extended Data Fig. 6e and 6f)."

- Extended Data Fig. 9. Panel A should replace Fig 3d in the main text.

We have done this.

No error bars for Panel C.

See comment above re. our ITC experiments.

As should the statements in the figure legend to the discussion in the maintext: “Increasing Lid domain flexibility by removing the disulfide bridge is linked to protonation and does not directly change affinity towards glucose (as seen for low pH). However, at higher pH, Asp42 does not become protonated easily, leading to lower turnover and lower glucose affinity.”

We have moved this text to the main text, which now reads (line 183):

“Increasing the Lid domain flexibility by removing the disulfide bridge is linked to protonation and does not directly change affinity towards glucose (as seen at low pH). However, at higher pH, Asp42 does not become protonated easily in the Lid mutant, leading to lower turnover and a threefold higher K_m (Extended Data fig 9c). ”

- Line 41 “apoplastic sugar depletion”: add 1 more sentence to clarify the phenomena.

We have changed the sentence to read (line 42)

“Furthermore, apoplastic sugar depletion through STPs, where sugar is removed from the extracellular space, has recently been identified as a defense strategy against microbial infection including rust and powdery mildew^{3-5,9-11}. By removing apoplastic sugar, the plant restricts the amount of nutrition available to the pathogen.”

- Line 47-51: the intro would read better if this section came after “...throughout the plant kingdom” in line 45.

We have move this section as suggested.

- Line 54 “...pH is alkalized”: add 1 more sentence on what this alkalization accomplishes.

We have added a sentence as suggested (line 56):

“STPs face the apoplastic space where pH is alkalized as a central stress response to e.g. microbial infection, drought and high salinity^{9-11,26,27}. The functional effects of this extracellular alkalization are not well understood, but the phenomenon, which can last from hours to days, is thought to form part of a central plant response to stressors²⁶. ”

- Line 58: In addition to Ext. Fig. 2, prepare an Ext. Table. 2 with % identity table (similar to Ext. Table 1)

We have added such a table as Extended Data Table 2.

- Line 60-63: This should be in the introduction.

We have moved the sentence up to the introduction of STPs (line 60).

- Line 88 “...C6...”: mention that this is the carbon coordinate of glucose; apply to other places in the text where only C4, C5 etc, is used.

We have added this and the sentence now reads (line 100):

“Asn332(M10) is in contact with the hydroxyl group of glucose carbon 6 (C6) while a hydrogen bond network made through Gln295(M7), Gln296(M7), Asn301(M7), Asn433(M11), Thr437(M11), Trp410(M10) and the main chain carbonyl of Gly406(M10) and water, mediate the contact to the C1-C4 hydroxyl groups of glucose. ”

- Line 98: Reference figure 2D at the end of the sentence in this line.

We unfortunately cannot follow the reviewers rationale in this particular case, as that sentence does not refer to fig 2d but is linked to the previous sentence that related to the growth complementation assay, which is referenced.

- Line 115-116 “Interestingly, rust resistance in wheat can be conferred by the Lr67 gene (encoding an STP) which contains a mutation at exactly the equivalent of the Arg142 position”.: So a positive trait emerges as a result of a mutation that should abolish proper STP function? Explain.

Some of the key mutants from the Moore et al paper have now been confirmed in barley (Milne et al *Plant Phys* 2018). We have added this new reference.

In the Moore et al paper it is suggested that “*LR67res may cause reduced hexose transport through a dominant-negative interference mechanism by forming inactive heteromultimeric protein complexes.*”. Ie. the protein is inactive, and forces some other protein (likely another STP) to be inactive as well.

There is a long jump from structural studies of monodisperse isolated protein functionality to the physiological effect on whole organisms, and this is a good example of that. Perhaps the discussion would be better suited to a more in-depth analysis in a review, but we have added a few sentences on the link to the Moore paper as we find it so interesting. In the context of our new findings, our key point is solely that the regions and residues we now suggest a mechanistic function for, also has a direct effect on physiology. The precise mechanisms by which this happens is still not elucidated.

The sentence has been rewritten (line 144):

“Interestingly, wheat and barley resistance towards fungal pathogens can be pinpointed to a glycine-to-arginine mutation in exactly this part of the N domain in a wheat sugar transporter (Lr67res) and the barley transporter HvSTP13, highlighting the importance of flexibility and charge distribution in this region^{4,5}.”

- Line 120 “This is consistent with the low pH of the crystallization...”: Pka of side chain Asp is 3.9 and your buffer 4.5. You need to make a point about the local amino acid profile and its influence on pKas of amino acids such as your Asp, to convince readers that instead of being mainly deprotonated, your Asp is protonated.

This is a good point. The effect on pKa when a charged residue is buried in a hydrophobic environment is discussed multiple times in the literature. It is not uncommon for the pKa values to shift by over 4 pH units towards the neutral form compared to in a pure water solution. For instance see an interesting study by Panahi et al *J Phys Chem* 119 (2015) (PMC4404502). Here they get a pKa value of 9.4 for Asp when embedded in a hydrophobic environment.

Programs exist that try to predict the actual pKa values of specific side chains in a given protein structure. However these programs are optimized for solvent accessible positions and will give incorrect results for membrane buried charged residues.

We have explained this in the manuscript (line 151):

“This is consistent with the low pH of the crystallization condition of 4.5, given that the Asp42 pKa would be expected to increase when removed from the positive charge of Arg142 while buried in a hydrophobic environment^{35,37}. It also matches previous observations from other proton symporters and proton pumps^{35,36,38}.”

- Line 130 “(Fig. 2a and Fig. 3a).”: These features (Phe 79...) are not visible on these figures, use Fig. 3b instead (Add trp 202 which is missing). Or make a new figure based on 3b.

We have labeled the aromatic residues on figure 2a, and changed the reference from fig. 2a+3a to fig. 2a+3b.

- Line 132. From your figures, it looks like Asp42 could be accessed by water. Explain.

This is correct. In the structure, the proton site is still linked to the extracellular side by a water accessible cavity. The structure is thus substrate occluded but still proton accessible. This is one of the reasons we call it a “substrate bound outward occluded state”. The open/occluded terminology in the transport field is often somewhat inaccurate as it only refers to the state with respect to the main substrate (here sugar), while the co-substrate (here protons) are ignored. This is mostly to keep the terminology simple. The current paradigm of active transport proposes that the protein will go from this observed state to a 'fully occluded state', before going to an inward facing conformation. The presented structure is thus “sugar-substrate outward occluded and proton-substrate outward-open”, and appear to capture a state between outward open and fully occluded. This type of conformational state has also been observed in several other Major facilitator structures. We mention the access in line 132.

- Line 138: “...confers pathogen resistance...”: Why does the mutation help?

This sentence contained an error and has been rewritten. We do not yet understand the precise mechanism by which this physiological change is mediated (see also the discussion with reviewer #4 point 4). The sentence now reads (line 181):

“Mutating the equivalent of Cys77 in the wheat gene Lr67 reintroduces pathogen susceptibility to the resistant gene-version (Lr67res)⁴, highlighting the impact of this cysteine.”

- Line 146: Described rocker-switch model if you want to challenge it (briefly).

We don't find that a description of the rocker-switch model is suitable for the manuscript as it has been described in great detail elsewhere, and in the end how STPs fit into this very broad transport model is not part of the discussed findings. We also agree that the sentence in general was overstated and we have thus removed parts to tone it down.

The sentence now reads (line 193):

“The Lid domain locks the two transmembrane domains together at the extracellular side via the disulfide bridge.”

- Line 149: Give pH range for the transport.

We have done this (line 198):

“The aromatic cluster of the Lid domain isolate the protonation site and enable efficient transport at the physiological pH of the apoplast (around pH 5-6)”

- Line 169-170: How do your experiments ‘shed new light on sugar transport’?

On the basis of the presented structure and characterization we now much better understand how sugar substrates are recognized by STPs, and we can, with a fairly simple mechanistic model involving the helix M1b and a proton donor/acceptor and a single hydrophobic residue (L43) explain how high substrate affinity is generated. In this sense we believe that we now have a better understanding of sugar transport than previously. The results likely are applicable beyond the kingdom of plants, as at least algae (HUP1/HUP2 example) also appear to utilize a similar mechanism.

The sentence reads (line 217):

“It sheds new light on sugar recognition and in particular explain how high affinity sugar transport can be generated, in a process that is essential to all plant life.”

Reviewer #4 (Remarks to the Author):

The manuscript by Paulsen et al. describes the first structure of a member of the important MST sugar transporter family from plants named STP10 at a resolution of 2.4 Å. Given the importance of STPs and the current discourse regarding their role in pathogen susceptibility this work is of high relevance and lays the basis for a better understanding of the currently difficult to understand phenomena described for example by Moore et al or Yamada et al. Moreover, the transporters in this family play crucial roles in many aspects of plant physiology, equal in importance as human GLUTs. The gained insights into the STP structure are also relevant in the context of understanding how some of these transporters function as uniporters (GLUTs), others as proton symporters (STPs). STPs are phylogenetically related to the human GLUTs and the structure solved here now enables us to compare structures, functions and regulation. Exciting new insights include the identification of a putative binding site for the substrate, the presence of an unusual lid with a disulfide bridge, and the conserved presence of an intracellular ICH domain. In addition the authors test the role of the two lid Cys for activity. They observe that mutation of cys reduces activity. They predict residues that may be involved directly in proton coupling and show that mutation reduces activity. They find the protein in an outward facing occluded state and generate a model for the transport mechanism, however this model needs to be interpreted with maximal caution in the light of the unusual lid domain.

We thank the reviewer for the enthusiasm for the presented work.

The reviewer is not a structural biologist, thus can not judge the quality of the structure, however it appears important to carefully check the LID domain, since it is a unique feature. Also it should be noted that it is conceivable that the disulfide bridge is not always present in vivo.

We agree the Lid domain was an unexpected finding. However the structural quality is such that there can be no doubt about the domain fold nor the actual cys-cys bridge in the crystal. We discuss the disulfide bridge below in point 5, but in brief, we argue that it is highly unlikely that the disulfide would not be present *in vivo* due to the experimental setup, as well as the perfectly conserved sequence features of both cysteines and aromatics residues of the Lid domain throughout the plant kingdom.

There is quite some literature that could be integrated here to discuss the structure and the residues relevant for activity. Sauer had altered the substrate specificity of an STP. Tanner had characterized residues important for substrate recognition in the Chlorella homolog, which is called HUP1. An important manuscript to be mentioned in the context of the ICH domain is the loss of function caused by deletion of the highly conserved C-terminus (conservation of cytosolic C termini is an exception, AMTs have highly conserved C termini important for function, STPs apparently as well). Examples are Grassl et al. 2000, Will et al. 1998, Buettner Sauer 2000. Very interesting is for example the finding from Tanner that deletion of part of the C-terminus in HUP1, an ancestor of the STPs leads to loss of activity, a mutation that also leads to loss of resistance in STP13 (Moore). This should all be discussed in the light of the possible role of the ICH domain.

We highly appreciate the thorough knowledge of the literature the reviewer has shared with us here and elsewhere in the review!

We agree that there is a lot of interesting observations related to the ICH domain which clearly is involved in both regulating actual transport as well as trafficking of the STPs (and other SP proteins). Several interactions that appear to stabilize the outward facing conformation involves these residues. This has also been observed in other sugar transporters such as Xyle and discussed in detail for human GLUT1 (Deng et al Nature 2014). At the same time a recent paper (Yamada et al. Plos One 12 (2017)) observes an ER export motif (WxxHxxW) in IC5 (the C terminal of the protein) of STP1 and STP13 which could explain several earlier findings of loss of activity in the literature. This is exactly the motif that also leads to loss of function in the Grassl et al (2000) paper, and its role in trafficking explain the phenotype observed in the Moore (2015) paper. This motif is now highlighted in the sequence alignment of the manuscript (blue box). We have tried to integrate some of these observations in our main text, but we also find that much of this needs a careful review to do all the observations full justice, and while interesting, are beyond the scope of this manuscript.

The manuscript now includes (line 91):

“Exit towards the cytosol is blocked, facilitated by several strong interactions between the N and C domain, that has also been observed in other sugar transporters (Extended Data Fig. 5). At the cytosolic side, the ICH domain forms part of this interaction and contributes with several hydrogen bonds to a network that involved both the N, C and ICH domains. This network has strong similarity to the one described for Xyle and human GLUT1, and key residues have previously been implicated in both activity and trafficking of STPs^{3,4,28-30}. ”

The Will et al (1998) paper was very interesting to us (we had previously missed it), because it actually confirms our L43A and L43N mutation in the HUP1 case, demonstrating that our proposed model for high affinity appears to be conserved here as well.

We have included HUP1 in the alignment of STPs (Extended Data Fig 1), to help the reader orient themselves when comparing our results to earlier literature on HUPs.

HUP2 has high affinity due to I47 (The prior position is the key proton donor/acceptor D46, equivalent to our D42), which when mutated to N decreases affinity 20-fold. Reversely, HUP1 has N45 in the equivalent position, and here the replacement to isoleucine give a 20 fold higher affinity. This is exactly what we also observe for STP10, when compared to both Xyle and Glut3 that both contain a polar residue (Thr) in the position immediately following the key proton donor/acceptor Aspartate.

We have integrated this in the manuscript (line 160).

“ A corresponding mutant L43N which replaces the hydrophobic interaction surface to a polar one of similar size, also leads to a significant decrease (Km 149 μM), highlighting that the hydrophobic aspect of the interaction is a key contributor to affinity (Extended Data Fig. 9a). Supporting this key role of Leu43, a similar pattern is observed in related sugar transporters HUP1 and HUP2 from the algae Parachlorella kessleri, where mutating this position changes affinity 20-fold depending on side chain hydrophobicity (Extended Data Fig 1)³⁹. This M1b-linked mechanism to control affinity thus appear to be conserved not only in STPs but also in closely related protein families outside the plant kingdom (Extended Data Fig. 1). ”

Major comments:

1. The authors fail to cite key references regarding the role of STPs (Norholm 2006 role in cell death; Schofield 2009 nitrogen use efficiency and growth, Yamada et al 2016 Science STP1 and 13 and Pseudomonas, signaling, interaction with FLS2 and BAK1; Lemmonier 2014 Botrytis resistance which is exact opposite of what Moore finds in wheat).

In the original manuscript we had a proposed limitation on the number of references, making it challenging to do full justice to all prior work. Principally we completely agree, and we have added the references

mentioned above in their proper context. Yamada 2016 and Lemmonier 2014 was already cited, and we have added a sentence to include the two other references (line 46):

“STPs have also been implicated in nitrogen use and in programmed cell death.”

They also do not discuss homo-oligomerization at all, which Moore claims to be key to resistance in the dominant resistance seen in hexaploid wheat. It is interesting that also other structural analyses of for example GLUTs have avoided to discuss oligomerization – e.g. Carruthers had suggested that GLUTs function as dimers of dimers, in which two subunits always switch together in the antiparallel orientation, respectively. It appears that none of the structures here shows signs of di- or higher oligomeric states. Important to mention.

The discussion of higher oligomeric states in Major Facilitators (not solely sugar transporters), have a long history. All crystallographic structures (Except the plant nitrate transporter NRT1.1 (Parker et al Nature 507 (2014) and Sun et al Nature 507 (2014))) solved so far have been monomeric, limiting the insight we can have on the effect of higher oligomeric states on the mechanism of transport. All structural data so far on Major Facilitators support the concept of transport through the monomer.

Understanding oligomeric states will be key to understand important aspects of trafficking and regulation in many Major Facilitators, including STPs and GLUTs, and we know of several groups that are pursuing this question from a structural perspective using eg. cryo-EM methods.

We have added the sentence (line 74):

“The asymmetric unit comprises a single monomer with no higher oligomeric state observed, despite STPs possibly being oligomers in a physiological context⁴.”

2. The authors do not compare structures to GLUTs (R0 etc) and do not highlight possible similarity and possible major structural differences. At least in the supplement a figure that focuses on the binding site should be shown side by side with binding sites from other members of MFS sugar transporters. They should discuss binding site interactions in more detail (taking also Sauer and Tanner data into account).

We now provide directly the RMSD between all sugar transporter structures in the new Extended Data Table 2. We compare the binding site of STP10 with Xyle and Glut3 (only other sugar transporters where structural data is available in an outward facing conformation) in Extended Data fig 7. We have expanded this figure to better highlight differences and similarities as suggested. Binding site interactions are now discussed in more detail in several places as described below for pt.3 as well as above where the connection from STP10-L43 to HUP1/HUP2 key mutations from Will et al (1998) are now discussed.

3. There is no experimental validation of the structure by testing the role of proposed substrate binding sites or other important domains. This is very easy in this case by using at least qualitative data from yeast complementation.

We have made a series of mutations to test some of the polar and pi-stack interactions to the sugar binding site. These results are now presented as Extended data Fig. 6c and d. The experiments confirm our structural findings: The F401A mutant abolishes transport highlighting the key function of the pi-stacking for substrate recognition. As expected the removal of the polar interactions from Q295A, N301A and N332A does not abolish transport, but gives STP10 much lower affinity for the substrate. All 3 mutants have poor growth at low glucose (1.1 mM) compared to wt, but show improved growth compared to wt at higher concentrations of glucose (10+ mM)(Extended Data Fig 6c). To validate this, we analyzed Q295A in further detail as an example and calculated the Km which is increased to 877 uM (Extended Data Fig 6d). Besides this new data, we have presented mutations of D42, R142, L43, C77 and C449. All are mutations that are used to confirm

the key observations from the structural data. These results in combination with the overlap to human and bacterial sugar transporters for the actual substrate binding site (discussed above for pt 2) validate the presented structural data.

The text now read (line 114):

“Further growth competition assays confirm the interactions of the polar and CH- π interactions from the C domain (Extended Data Fig. 6c and 6d). The F401A mutant abolishes transport, highlighting the pivotal function of a CH- π interaction for protein-monosaccharide recognition³². As expected, removal of single polar interactions (Q295A, N301A and N332A) does not completely abolish transport, but appear to give STP10 much lower affinity for its substrate as demonstrated for Q295A (Extended Data Fig. 6c and 6d). All of these interactions between the C domain and the substrate are also found in bacterial and human sugar transporters (Extended Data Fig 7). ”

4. There is also a major error in the interpretation regarding the residues that are important for dominant rust resistance (LR67) in STP13. I admit I also had to read the paper multiple times and look at their background papers before this became fully clear.

We are immensely grateful for the reviewer to point out this error in our interpretation! We have changed the text as described below to correct this.

Note that Moore uses LR67 for both the susceptible and the resistant version and marks them res variants. Moore et al. describe that Gly144 and Val387 are key to resistance in wheat (although from their data still unclear whether both are required). Then they describe suppressor mutations that lead back to susceptibility! These include 8 mutations listed in the Moore paper.

Yes, we agree. To summarize the Moore et al paper as we understand it:

STP(LR67)-wt: wheat susceptible to rust.

STP(LR67)-G144R/V387L: wheat resistant to rust. Proposed cause: “*LR67res may cause reduced hexose transport through a dominant-negative interference mechanism by forming inactive heteromultimeric protein complexes.*”. I.e. the protein is inactive, and forces some other protein (likely another STP) to be inactive as well.

STP(LR67)-G144R/V387L-[one-of-the-8-mutants]: Wheat now again susceptible to rust. The simplest explanation is that the additional mutation inhibits the formation of the heterodimers, thus reestablishing susceptibility by allowing the other partner of the heterodimer to become active again, and compensate for the loss of STP(LR67).

This would fit the general model where STPs are involved in pathogen resistance by removing sugar from the apoplastic space as reported in eg. Yamada 2016 and Lemonnier 2014. (but other models are possible as well).

Some of the key mutants from the Moore et al paper have now been confirmed in barley (Milne et al *Plant Phys* 2018). We have added this new reference.

They state that Gly144 in the susceptible version (the quasi wild type) is conserved in all STP13s. The resistant version contains an Arg in that position, creating a double R at the position in the membrane with possibly dramatic effects on the transmembrane helix. Thus the Arg142 discussed for STP10 in this manuscript is right next to the equivalent of the Gly144 Arg described by Moore.

Correct. We apologize for the confusion. We have adjusted our text (line 144):

“ Interestingly, wheat and barley resistance towards fungal pathogens can be pinpointed to a glycine-to-arginine mutation in exactly this part of the N domain in a wheat sugar transporter (Lr67res) and the barley transporter HvSTP13, highlighting the importance of flexibility and charge distribution in this region^{4,5}. ”

Similarly, the statement that the equivalent of Cys77 in wheat STP13 is relevant to resistance is not correct.

We have corrected our text. It now reads (line 181):

“Mutating the equivalent of Cys77 in the wheat gene Lr67 reintroduces pathogen susceptibility to the resistant gene-version (Lr67res)⁴, highlighting the impact of this cysteine. ”

The suppressor screen identified mutations that suppress resistance, and the most likely explanation is that all suppressor mutations lead to a loss of function, retaining the protein but eliminating the activity, which leads to resistance. This needs to be clarified and corrected throughout the text and discussed in more detail.

There is a long jump from structural studies of monodisperse isolated protein to the physiological effect on whole organisms, and this is a good example of that. Perhaps the discussion would be better suited to a more in-depth analysis in a review, but we have added a few sentences on the link to the Moore paper as we find it so interesting. In the context of our new findings, our key point is solely that the regions and residues we now suggest a mechanistic function for also has a direct effect on physiology. The precise mechanisms by which this happens is still not elucidated.

5. The predicted lid and the disulfide bridge in the predicted lid could potentially be very exciting. The authors tested the importance of the Cys residues for activity. They observe a reduction in activity, but while interesting, these experiments do not address the key question – what is the lids role and is it regulated.

We hypothesize that the key role of the Lid domain is to create a protected environment for the proton donor/acceptor residues Asp42 and Arg142. We do not currently address if the Lid is part of a regulatory mechanism. While that might be a possibility *in vivo*, we would need to identify regulatory partners or pathways to fit this speculative model, and that is beyond the scope of the presented work.

They rather make strong statements that are not warranted by any data, for example that the finding of the lid challenges the rocker switch model.

We have now removed this statement from the conclusion, as it is not essential to our main findings.

It is important to test redox dependence of the activity, as shown to be important for other transporters, for example the regulation of the yeast calcium channel by redox regulation (Cch1p glutathionylation).

The yeast calcium channel is an interesting case study on redox regulation. That process is happening inside the cytoplasm. The cytoplasm is a reduced environment. If the cell is stressed the cytoplasm can turn into a more oxidative environment (a well-known and characterized response) which is sensed by eg. the calcium channel.

We are not aware of any mechanism by which the extracellular space (where the observed disulfide bridge is located) could change to a reducing environment. Under normal conditions the extracellular space must be considered an oxidative environment, and since the two cysteines are very close together in space they are very likely to form a disulfide bond. The default assumption given the solved structure we present, must be that STP10 has a disulfide bridge *in vivo*. We are not aware of any example from the structural literature

where an observed intramolecular cys-bridge involving cys-residues located in an extracellular environment turned out to be non-physiological.

Perhaps the cys-bridge can be removed in extreme environmental (anaerobic?) cases, but it is well beyond the scope of this paper to speculate on such a physiological mechanism in the plant apoplast. Such a mechanism would be extremely interesting and something that might be worth considering for future experiments.

To further strengthen our observation of the cys-bridge, we have done an *in vivo* uptake experiment in yeast where we add reducing agent. This experiment is now shown in Extended Data Fig 9b. The data shown that while uptake with the cys77Ala mutant is unaffected by high levels of reducing agent, activity in the wt form is reduced to ~51% when exposed to reducing agent. This further supports that the disulfide bridge is present *in vivo*.

The text now reads (line 177):

“However, more detailed investigation shows that both the Cys77Ala and Cys449Ala mutant becomes increasingly sensitive to alkaline pH and can only function fully at acidic pH (pH < 5) (Fig. 3d and Extended Data Fig. 9). In confirmation of this, only the wt protein is sensitive to reducing agents in an in vivo uptake assay, indicating that the disulfide bridge is present and activity is lowered when the bridge is reduced (Extended Data Fig. 9b).”

Is it not likely that this disulfide bridge is biologically and structurally relevant. Could it be that this helped to lock the structure into the observed conformation and occurred during purification? Can it be reduced in the structure?

The disulfide bridge is clearly present in the structure. The purification is done in the presence of 0.5 mM TCEP and the likelihood that the cysteines do not form a disulfide bridge during oxidative growth, but then spontaneously form during purification in a reducing environment is very low.

Furthermore that observation that the two cysteines in question are perfectly conserved in all STP sequences (over 3000 plant STP sequences analyzed by Consurf), as well as in the HUP1 case, clearly point to some function. Since both cysteines are located on extracellular loops, and do not form part of binding sites or secondary structural elements, it is difficult to imagine what other key function they could have. We cannot exclude a regulatory function, but we currently have no evidence that the cysteines nor the cys-bridge would be regulatory, and suggesting such a function at this point would be too speculative.

Minor comments

1. The title is not informative and unintentionally misleading. It is not the authors fault, since STP stands for Sugar Transport Protein, but for the broad audience misleading and unclear. Important to change the title to make clear it's a plant STP or MST related to GLUTs.

We agree the initial title was too generic. We have changed the title to:

“Crystal structure of the plant proton/sugar symporter STP10 illuminates the mechanism behind sugar uptake in the Monosaccharide Transporter Superfamily”

2. Summary: the study does not identify the proton donor acceptor pair, it provides circumstantial evidence that these might be the residues. Needs to be tuned down

We have toned this down, and use the word “suggest” instead.

3. STPs are not active, they are secondary active.

We naturally agree that STPs are secondary active transporters. However we cannot find any place in the text where it is incorrectly stated that they are something else than that. We are very happy to fix this mistake in the manuscript if pointed in the right direction.

We found one place where it states : "*STP10 is active and display a high affinity transport of glucose* ”.

While this might not be the place the reviewer had in mind, we have adjusted this sentence as we found it to be unclear in the context what is meant. It now reads (line 105):

“ STP10 display high affinity transport of glucose ... ”

4. The statement that STP10 has a very high selectivity for three substrates is odd, and not warranted by the data, since only a limited number of soluble sugars and glycosides was tested. SUTs have been found to transport for example esculin, fraxin and helicin.

We agree that our original language was not precise. We have adjusted this throughout the text. Eg. We have changed line 61 to read:

“Found in growing pollen tubes, it is a proton driven symporter that displays low μ M range affinity for glucose and can transport glucose, galactose and mannose²⁵”

5. High glucose concentrations (high is relative) do not have these effects in all experiments and plants. And the sentence that states this is odd since it states sugars stunt plants as do STPs...STPs are just transporters. Do you mean that the plants become more sensitive to glucose in particular when STPX is overexpressed?

We apologize for the confusion. Our initial sentence was poorly formulated, and actually only aimed at discussing growth of yeast in a growth complementation assay. We have rewritten the sentence (line 110), shown here in context:

“Furthermore, growth complementation assays show uptake of glucose at lower sugar concentration (\sim 1 mM), and for both mannose and fructose at higher sugar concentrations (>10 mM) (Extended Data Fig. 6b). High glucose concentrations (>10 mM) appear to inhibit yeast growth as also reported previously for other STPs in growth complementation assays^{27,31}. “

6. This effect is confirmed – reference is unclear and confirmed is too strong, replace by supported

We have changed this as suggested.

7. The statement that the large tolerance for pH created by the lid is directly related to STPs physiological function is too strong and unclear, different members have different functions, and the pH profile of measured activity is created by a combination of the pH profile of the protein itself and the proton coupling (and if tested in yeast influenced by the acidification by the yeast when it receives sugars).

We have toned this down. The sentence now reads (line 206):

“ The broad pH tolerance created by the Lid domain can be related to the physiological function of STPs, where STP activity is preserved during stress-induced alkalization of the apoplast. ”

Reviewers' Comments:

Reviewer #1:

Remarks to the Author:

My comments have been adequately addressed. This study is a very nice and important contribution to the field.

Reviewer #3:

Remarks to the Author:

No further comments

Reviewer #4:

Remarks to the Author:

The authors have addressed almost all comments made by the reviewer.

There are some remaining aspects that the reviewer recommends to address by altering the text.

While the reviewer is aware that publication in top journals sometimes requires highlighting the novelty, it is important at this point to tune down some interpretations, without impacting the importance of this work. Lets assume that redox can change and affect the disulfide bridge formation. And lets assume that the residues implicated in proton coupling here are not key to the process. If true, then it would be better at this point to tune down the conclusions, and present alternative hypotheses. STP1 is a plant protein, and it is conceivable that redox outside the cells (e.g. at the root surface in soil, but also in the cell wall between cells in various organs may change. For example, Graff et al (<https://doi.org/10.1093/jxb/erq379>) had found indications that cysteine in ammonium transporters may play a role in oligomerization, and thereby regulation of activity.

Thus addition of a sentence stating that it is also conceivable, that under certain conditions, the lid may be affected. Of course its also conceivable, that if STPs form dimers, then the bridge may occur between subunits. I recommend to add one or two sentences, and also tune down the conclusions around proton coupling.

REVIEWERS' COMMENTS:

Reviewer #1 (Remarks to the Author):

My comments have been adequately addressed. This study is a very nice and important contribution to the field.

We thank all the reviewers for their valuable input to our manuscript.

Reviewer #3 (Remarks to the Author):

No further comments

We thank all the reviewers for their valuable input to our manuscript.

Reviewer #4 (Remarks to the Author):

The authors have addressed almost all comments made by the reviewer.

We thank all the reviewers for their valuable input to our manuscript.

There are some remaining aspects that the reviewer recommends to address by altering the text. While the reviewer is aware that publication in top journals sometimes requires highlighting the novelty, it is important at this point to tune down some interpretations, without impacting the importance of this work. Lets assume that redox can change and affect the disulfide bridge formation. And lets assume that the residues implicated in proton coupling here are not key to the process. If true, then it would be better at this point to tune down the conclusions, and present alternative hypotheses. STP1 is a plant protein, and it is conceivable that redox outside the cells (e.g. at the root surface in soil, but also in the cell wall between cells in various organs may change. For example, Graff et al (<https://doi.org/10.1093/jxb/erq379>) had found indications that cysteine in ammonium transporters may play a role in oligomerization, and thereby regulation of activity. Thus addition of a sentence stating that it is also conceivable, that under certain conditions, the lid may be affected. Of course its also conceivable, that if STPs form dimers, then the bridge may occur between subunits.

I recommend to add one or two sentences, and also tune down the conclusions around proton coupling.

We have tuned down our conclusion.